# VisualScore: Learning Holistic Visual Quality Scores via Multi-Task Reasoning

**Yiting Lu** [* 1 2]  **Fengbin Guan** [* 1]  **Yixin Gao** [* 1]  **Yan Zhong** [2 3]  **Xinge Peng** [1]  **Jiakang Yuan** [4 5]  **Yihao Liu** [4]
**Bo Zhang** [4]  **Xin Li** [1]  **Zhibo Chen** [1]  **Weisi Lin** [2]

## Abstract

Image quality assessment (IQA) is inherently multi-dimensional, yet existing reward models are typically limited to a single task and become unstable when extended to multi-task settings. In particular, heterogeneous reward scales and variances across tasks can lead to conflicting optimization signals during reinforcement learning. We propose VisualScore, a unified visual evaluation framework that formulates multi-task IQA as structured, task-aware reasoning followed by continuous reward optimization. VisualScore produces interpretable rationales together with scalar quality scores under explicit evaluation principles. We construct a reasoning-enhanced reward modeling dataset via rejection sampling and initialize the model through supervised fine-tuning. VisualScore is then optimized with Group Relative Policy Optimization (GRPO) using a Gaussian-based continuous reward. To address multi-task reward conflicts and stabilize training, we introduce standard deviation filtering and entropy gating to normalize task-wise reward signals and suppress noisy updates. Experiments on technical quality, aesthetic quality, and text–image alignment show that VisualScore improves robustness, generalization, and interpretability, and can effectively guide text-to-image generation at test time without retraining. Our code is available at https://github.com/yeppp27/VisualScore.

## 1. Introduction

The rapid growth of visual data, including user-generated content (UGC) and AI-generated content (AIGC), presents new challenges for image quality assessment (IQA) across

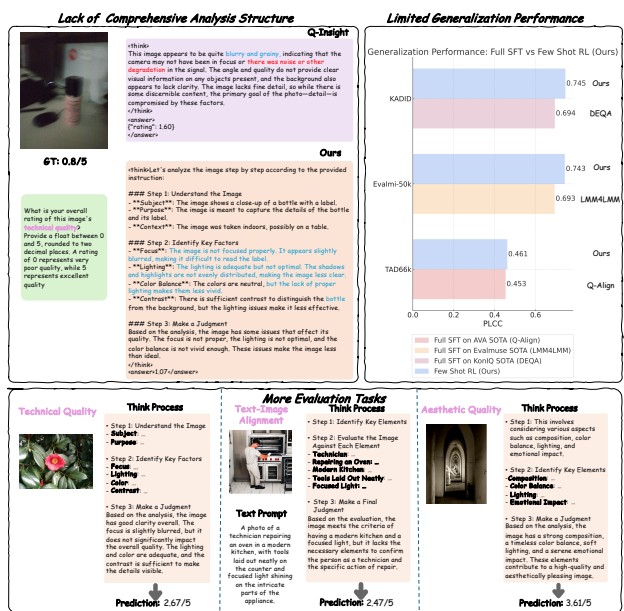

*Figure 1.* Overview of the challenges in multimodal image quality assessment and the proposed VisualScore framework. **Left**: Existing methods lack a structured reasoning process, often yielding incomplete analysis, whereas VisualScore introduces a step-by-step, interpretable reasoning structure. **Right**: VisualScore achieves improved generalization performance, particularly with few-shot data through reinforcement learning (RL), outperforming existing SOTA models across multiple benchmarks. **Bottom**: VisualScore supports multi-dimensional evaluation—technical quality, aesthetic appeal, and text-image alignment—each guided by a transparent think process and yielding final quality predictions.

diverse domains and tasks (Agnolucci et al., 2024; Min et al., 2024; Sun et al., 2024; Li et al., 2024c; Yuan et al., 2024; Yu et al., 2024a; Fang et al., 2024; Yu et al., 2024b). Traditional IQA approaches, which rely on hand-crafted features (Mittal et al., 2012b;a) or neural networks (Talebi & Milanfar, 2018; Su et al., 2020; Network, 2022; Ke et al., 2021; Wang et al., 2023; Yang et al., 2022; Zhong et al., 2025; 2024; Guan et al., 2025a; Peng et al.) trained on synthetic or authentically distorted image datasets, often exhibit limited generalization to new data. Moreover, these approaches typically output a single scalar score, offering little insight into the underlying reasons for the quality judgment and lacking interpretability. As IQA increasingly plays a critical role in guiding image post-processing (e.g., super-resolution,

---
[*]Equal contribution  [1]University of Science and Technology of China [2]Nanyang Technological University [3]Peking University [4]Shanghai Artificial Intelligence Laboratory [5]Fudan University. Correspondence to: Weisi Lin <wslin@ntu.edu.sg>, Zhibo Chen <chenzhibo@ustc.edu.cn>, Xin Li <xin.li@ustc.edu.cn>.

*Proceedings of the 43rd International Conference on Machine Learning*, Seoul, South Korea. PMLR 306, 2026. Copyright 2026 by the author(s).

restoration) (Jiang et al., 2025; Li et al., 2025a; Chen et al., 2024b), image compression (Wang et al., 2025b; Li et al., 2024d; Guan et al., 2025b;c), and serving as a reward model for preference optimization in text-to-image (T2I) generation (Luo et al., 2025; Wang et al., 2025d; Xu et al., 2024; Ma et al., 2025; Chen et al., 2024c; Xie et al., 2025a; Qin et al., 2025; Lu et al., 2024; Luo et al., 2026; Liang et al., 2026a;b), the demand for robust and explainable reward models through all-encompassing quality assessment has become more pressing. These conventional methods are typically constrained by the domain-specific characteristics of their training data, limiting their applicability to the diverse, high-variance visual content encountered in modern applications. The recent development of vision-language models, such as CLIP (Sun et al., 2023; Radford et al., 2021) and BLIP (Li et al., 2022; 2023), offers new opportunities for IQA. By leveraging rich vision-language pretraining, these models enable more robust quality evaluation across a wide range of content types (Wang et al., 2023; Zhang et al., 2023). Recent advances in quality-aware multimodal large language models (MLLMs) (Wu et al., 2024b; Zhang et al., 2024b; Chen et al., 2024a; Wu et al., 2024a; Jia et al., 2024; 2025; Wu et al., 2023a) have further expanded the landscape of IQA (Zhang et al., 2025). Several approaches explicitly design prompts or conduct supervised fine-tuning (SFT) to inject quality sensitivity into MLLMs. For instance, Q-Instruct (Wu et al., 2024b) and Co-Instruct (Wu et al., 2024c) adopt instruction tuning on diverse quality-related question answering and caption tasks, while Q-Align (Wu et al., 2023a), Q-Boost (Zhang et al., 2024a) and DeQA (You et al., 2025a) focus on quality scoring tasks. Other works such as DepictQA (You et al., 2024) and DepictQA-v2 (You et al., 2025b) introduce text reasoning ability for full-reference image quality assessment, and Q-Ground (Chen et al., 2024a) leverages quality-grounded visual grounding to enhance localization of quality issues. Despite these efforts, current quality-aware MLLMs typically follow fixed reasoning patterns dictated by curated datasets and instructions, limiting their flexibility and depth in complex assessments. Specifically, existing methods are often restricted to a single evaluation task and lack structured chain-of-thought reasoning: for example, Q-Align (Wu et al., 2023a) focuses on single-task scoring without interpretable reasoning chains, while Q-Insight (Li et al., 2025c), though supporting reasoning, does not cover text–image alignment evaluation. This narrow task scope prevents these models from serving as unified reward models across heterogeneous quality dimensions. Even cutting-edge models like Qwen2-VL (Wang et al., 2024) and Qwen2.5-VL (Bai et al., 2025), while demonstrating strong general perception capabilities, remain relatively weak in fine-grained low-level quality perception (e.g., subtle distortions, compression artifacts) due to their broad-domain training objectives that do not specifically target quality-sensitive features.

To address these challenges, we introduce VisualScore, a unified reward model for multi-task image quality assessment that enhances multimodal reasoning through structured and interpretable evaluation across three representative quality dimensions: technical quality (low-level distortions), text–image alignment (semantic consistency), and aesthetic quality (subjective visual appeal), as illustrated in Fig. 1. By enforcing a shared reasoning structure across heterogeneous tasks, VisualScore improves generalization and enables consistent quality assessment under a unified evaluation paradigm.

To equip Multimodal Large Language Models (MLLMs) with robust quality-aware reasoning capabilities across diverse tasks, we design a two-stage training framework. In the first stage, we perform cold-start fine-tuning with rejective sampling to teach implicit question analysis and explicit reasoning structures based on task-specific evaluation plans. In the second stage, we apply unified reinforcement learning with Group Relative Policy Optimization (GRPO) to further explore and refine reasoning pathways for continuous score prediction. During reinforcement tuning, we adopt a Gaussian-shaped reward function to reflect the continuous nature of image quality scores and to provide smoother optimization signals than binary or preference-based rewards. However, in the multi-task setting, heterogeneous reward distributions across tasks introduce scale and variance mismatches, which can lead to unstable policy updates and task imbalance during reinforcement learning. To address this issue, we introduce a stage-wise stabilization strategy. Specifically, we employ entropy gating to prevent premature convergence on easier tasks and encourage balanced exploration of reasoning pathways across tasks. In parallel, we apply standard deviation (STD)-guided filtering to suppress task-specific low-variance signals that would otherwise dominate gradient updates and cause batch-level imbalance across tasks. Together, these mechanisms act as task-level normalization components, aligning heterogeneous reward signals and stabilizing multi-task reinforcement learning. This design significantly improves training stability and generalization, particularly on challenging out-of-distribution benchmarks.

- **VisualScore: Structured Multi-Task Quality Reasoning.** We propose VisualScore, a unified reward modeling framework that formulates image quality assessment as structured, task-aware reasoning across technical quality, aesthetic quality, and text–image alignment, enabling interpretable and comprehensive evaluation.
- **Two-Stage Training for Robust Quality-Aware Reasoning.** We design a two-stage training pipeline that combines cold-start rejective sampling fine-tuning with reinforcement learning using Group Relative Policy Optimization (GRPO), allowing MLLMs to learn both explicit reasoning structures and accurate continuous score prediction.

- **Task-Level Stabilization for Multi-Task Reinforcement Learning.** To address reward conflicts arising from heterogeneous scales and variances across tasks, we introduce entropy gating and standard deviation (STD) filtering as task-level stabilization mechanisms, which align reward signals and ensure stable multi-task reinforcement learning.
- **Generalizable Evaluation and Test-Time Guidance.** VisualScore achieves strong performance across diverse image quality benchmarks and can further serve as an interpretable reward function to guide text-to-image generation at test time, improving alignment and compositional fidelity without additional training.

## 2. Related Work

### 2.1. MLLMs

Recent progress in multimodal large language models (MLLMs) (Wang et al., 2024; Bai et al., 2025; Team et al., 2025; Zhu et al., 2025) has advanced along two major directions. The first focuses on developing general-purpose, state-of-the-art MLLMs such as Qwen2-VL (Wang et al., 2024), Qwen2.5-VL (Bai et al., 2025), InternVL3 (Zhu et al., 2025), Pixtral (Agrawal et al., 2024), and Kimi-VL (Team et al., 2025). These models are pre-trained on large-scale multimodal corpora and further refined through long-context supervised fine-tuning, achieving strong performance across diverse vision-language tasks and approaching proprietary systems such as GPT-4o. The second direction emphasizes enhancing visual reasoning abilities through post-training with high-quality chain-of-thought (CoT) supervised fine-tuning (SFT). For instance, Mulberry (Yao et al., 2024) employs Monte Carlo Tree Search (MCTS) with multiple MLLMs to construct long-horizon reasoning trajectories, which are then used to train reasoning-augmented models through SFT. Similarly, Mammoth-VL (Guo et al., 2024) leverages a 72B-parameter MLLM to rewrite reasoning chains for visual question answering tasks, which are subsequently used to fine-tune models.

### 2.2. MLLMs for IQA

Recent works have significantly advanced the fine-tuning and alignment of open-source MLLMs for IQA by introducing innovative training paradigms that balance accuracy and interpretability. Q-Instruct (Wu et al., 2024b) leveraged a novel dataset of human-authored low-level quality descriptions to instruction-tune MLLMs, markedly improving their ability to assess fine-grained distortions. In parallel, Q-Align (Wu et al., 2023a) reframed quality prediction as a classification task over discrete text-defined rating levels, aligning model outputs with human subjective rating categories for more calibrated scoring. To incorporate relative comparisons, Compare2Score (Zhu et al., 2024) trained on pairwise image comparisons and proposed a soft-anchor inference mechanism that compares test images against

anchor images to infer continuous quality scores, effectively integrating diverse IQA datasets and enhancing cross-domain robustness. On the interpretability front, DepictQA (You et al., 2024) enabled free-form, language-based quality assessment by prompting models to generate detailed descriptions of image artifacts and comparative judgments, while Grounding-IQA (Chen et al., 2024d) further introduced spatial grounding, requiring the model to localize specific regions causing quality degradation via referring expressions, thus achieving fine-grained quality analysis. More recently, reinforcement learning has been utilized to jointly optimize scoring and reasoning capabilities. Q-Insight (Li et al., 2025c) employed a Grouped Relative Policy Optimization (GRPO) framework to refine both score regression and degradation reasoning using limited human feedback, demonstrating improved accuracy and zero-shot reasoning generalization. Similarly, VisualQuality-R1 (Wu et al., 2025) adopted a reinforcement learning-to-rank approach that shifts training from absolute scoring to relative pairwise ranking with continuous rewards, yielding superior generalization across distortions and the ability to produce human-aligned quality explanations. Moreover, Q-Ponder (Cai et al., 2025) unified score and explanation alignment in a two-stage pipeline.

## 3. Method

We propose VisualScore, a reward model designed as a structured reasoning framework for all-encompassing image quality assessment (IQA) inspired by how expert judges evaluate images. Just as a photography judge first understands the evaluation goal, then identifies key dimensions such as composition, lighting, and artifacts to analyze individually before summarizing into an overall score, VisualScore decomposes the assessment into an explicit planning stage followed by step-by-step guided reasoning. This approach encourages the model to form interpretable, dimension-aware judgments, enabling more robust and explainable reward modeling across diverse evaluation tasks, including technical quality assessment, aesthetic quality assessment, and text-image alignment. In the following, we first describe how we construct a reasoning-enhanced dataset, then present our two-stage training pipeline: supervised fine-tuning on high-quality plan-reason trajectories, and post-training with Group Relative Policy Optimization (GRPO) using a continuous reward.

### 3.1. Cold-Start SFT Stage

**Plan-then-Reason Dataset Construction** The dataset construction follows a structured Plan-then-Reason paradigm for three multimodal image evaluation tasks: aesthetic quality, technical quality, and text–image alignment. Given a task-specific prompt, the MLLM first generates an explicit evaluation plan that decomposes the task into a set of concrete criteria, such as image clarity, color and contrast,

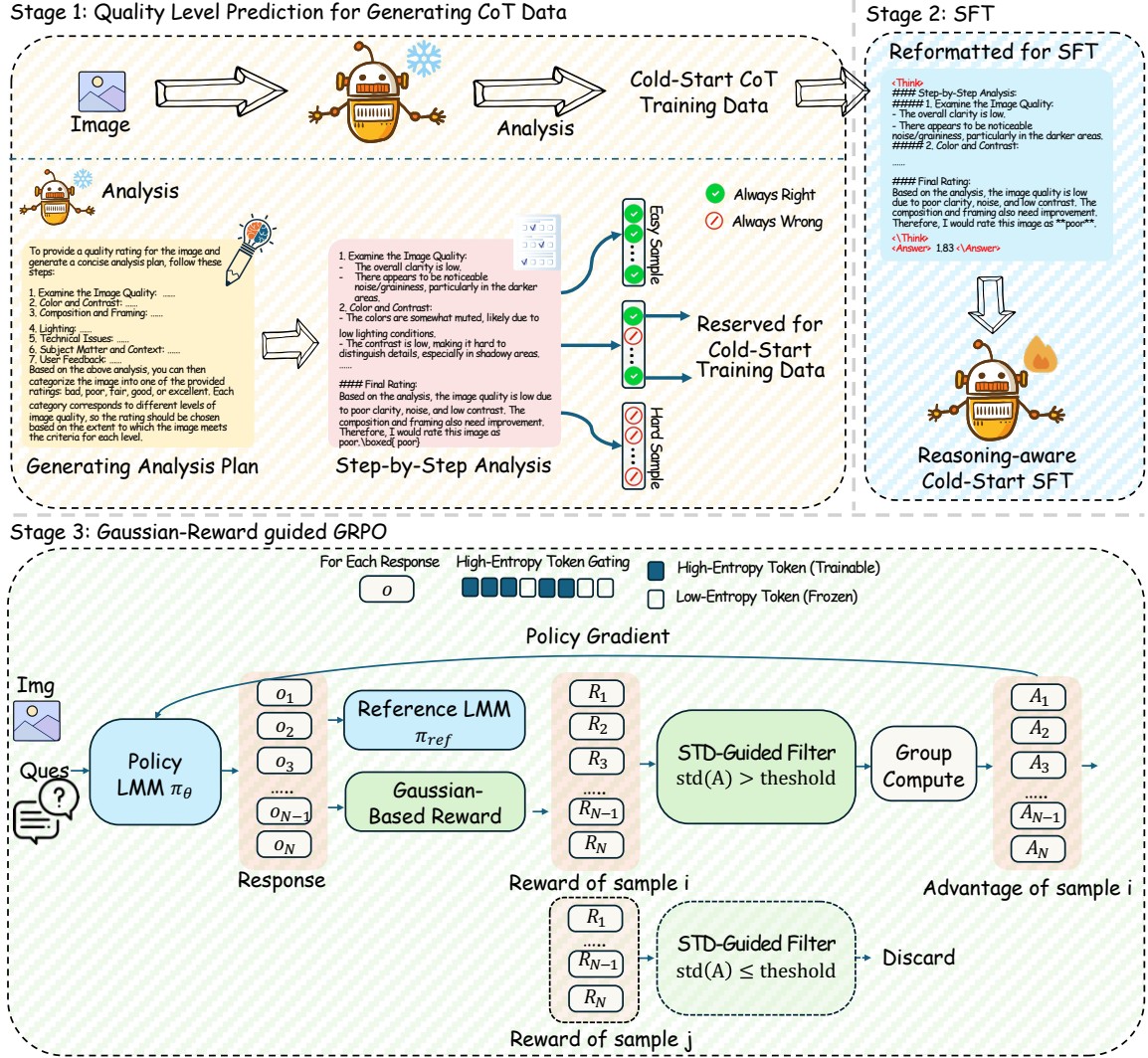

*Figure 2.* Overview of the VisualScore Framework. The framework consists of three stages: (1) Generating Chain-of-Thought (CoT) data with quality level prediction based on structured image analysis; (2) Reasoning-aware supervised fine-tuning (SFT) using reformatted CoT samples, emphasizing hard cases; (3) Gaussian-reward guided GRPO that applies a standard deviation-based filter and high-entropy token gating to optimize policy learning, improving reasoning robustness under rule-based supervision.

composition, lighting, and potential artifacts. This plan is then combined with the original prompt and the input image, and fed back into the MLLM to produce step-by-step Chain-of-Thought (CoT) reasoning. The resulting CoT not only supports the final quality score with interpretable decision steps, but also enforces consistent instruction-following behavior across different tasks.

**Rejective Sampling Finetuning** This method supports dataset creation across all three tasks by producing high-quality, traceable annotations that are used for supervised fine-tuning (SFT). Additionally, the process filters out both easy and hard examples, retaining the remaining samples for cold-start SFT training. This design mitigates the advantage vanishing issue during the subsequent reinforcement learning phase. Specifically, we first normalize all quality

scores to the range $[0, 5]$ and discretize them into five semantic levels (bad, poor, fair, good, excellent). For each sample, we run the model $N = 8$ times and compare the predicted quality-level word against the ground-truth level. If all $N$ predictions match the ground truth, the sample is classified as *easy*; if none match, it is classified as *hard*. Both categories are filtered out, and only the remaining intermediate-difficulty samples are retained for cold-start SFT, ensuring the model is trained on informative examples that provide meaningful learning signals. Before supervised training, we further refine the selected CoT trajectories using Gemini to correct visually grounded inconsistencies and align the reasoning steps with the corresponding visual evidence. Specifically, we prompt Gemini with the input image, the original CoT trajectory, and a verification instruction

that asks it to identify and correct any factual errors in the visual descriptions (e.g., incorrect color, misidentified objects, or hallucinated artifacts). This refinement step ensures that the reasoning chains are faithfully grounded in the actual visual content, improving the quality of SFT supervision. Finally, the reasoning-enhanced MLLM $\pi_{\text{reason}}$ is fine-tuned using supervised learning on the selected CoT trajectories. Notably, during the training phase, only the original question is retained while the previously generated analysis plan is removed. This design choice is intentional and serves to bypass the need for explicit plan generation during both the reinforcement learning and inference stages. By training the model solely on the original prompt paired with the reasoning-augmented response, the approach implicitly encourages the model to internalize effective planning and reasoning structures.

## 3.2. Reinforcement Finetuning with GRPO

In VisualScore, reinforcement learning is performed in a multi-task setting, where heterogeneous image quality objectives—technical fidelity, aesthetic preference, and text–image alignment—are jointly optimized under a unified policy. While this enables shared reasoning across tasks, it also introduces reward conflicts caused by task-wise differences in scale, variance, and learning difficulty, which can destabilize policy optimization if left unaddressed.

**Gaussian Reward** To provide a unified reward signal compatible with continuous score regression across tasks, we adopt a *Gaussian-shaped reward*:

$$R = \exp\left(-\frac{(\hat{s} - s^*)^2}{2\sigma^2}\right), \tag{1}$$

where $\hat{s}$ denotes the predicted score, $s^*$ is the ground-truth score, and $\sigma$ controls the sharpness of the reward decay. Compared to binary or preference-based rewards, this formulation produces smoother gradients and better aligns with the continuous nature of image quality assessment. More importantly, it enables consistent supervision across heterogeneous tasks, forming the basis for unified multi-task optimization.

**Standard Deviation-Guided Sample Filtering** Despite the smoothness of Gaussian rewards, near-identical rewards within a response group can lead to vanishing advantages and ineffective gradients. This effect is amplified in multi-task reinforcement learning, where tasks saturate at different rates. To mitigate this, we adopt a *standard deviation-guided filtering strategy* over sampled response groups. Let the training batch be divided into $M$ groups, each containing $K$ sampled responses with associated Gaussian rewards $\{r_j^{(i)}\}_{j=1}^K$ for group $i$. For each group $i$, we compute the intra-group reward standard deviation $\sigma^{(i)} = \text{std}\left(\{r_j^{(i)}\}_{j=1}^K\right)$, and retain the group for training only if $\sigma^{(i)} > \tau$, where $\tau = 0.001$. By focusing on high-variance groups, the model benefits from more informative feedback for policy gradient.

**Entropy Gating for Backward Policy Gradient** As training progresses, different tasks exhibit distinct entropy patterns along their chain-of-thought trajectories: simpler tasks quickly become over-confident, while harder tasks retain high uncertainty in intermediate reasoning steps. This mismatch leads to localized entropy collapse and unbalanced learning signals in multi-task reinforcement learning. To address this, following prior work on high-entropy token optimization (Wang et al., 2025c; Guo et al., 2025), we adopt a unified entropy gating mechanism that applies policy gradients only to high-entropy tokens. This design balances learning across tasks by emphasizing uncertain reasoning regions while avoiding redundant updates on over-confident task-specific trajectories. Crucially, the gating threshold is not a fixed absolute value but a relative quantile: we compute the top-$\rho$ quantile (with $\rho = 0.2$) over all token entropies within the current batch. Because the threshold adapts to each batch's entropy distribution, it naturally accommodates tasks with different average entropies. We modify the DAPO loss (Yu et al., 2025) (token-level loss) to apply policy gradients only on high-entropy tokens. $r_t^i(\theta)$ denotes the ratio between the current policy $\pi_\theta$ and the old policy $\pi_{\text{old}}$ for token $o_t^i$, i.e., $r_t^i(\theta) = \frac{\pi_\theta(o_t^i | q, o_{<t}^i)}{\pi_{\text{old}}(o_t^i | q, o_{<t}^i)}$, and $\hat{A}_t^i$ is the corresponding advantage estimate. $H_t^i$ is the entropy of the predicted token distribution, and $\tau_\rho^{\mathcal{B}}$ denotes the top-$\rho$ quantile threshold computed across all token entropies within the batch. The indicator function $\mathbb{I}(H_t^i \geq \tau_\rho^{\mathcal{B}})$ ensures that only high-entropy tokens contribute to the gradient, enabling the model to focus on uncertain and informative regions. Given a mini-batch $\mathcal{B}$ sampled from dataset $\mathcal{D}$, the loss is defined as:

$$\mathcal{J}(\theta) = \mathbb{E}_{\mathcal{B}}\left[\frac{1}{\sum_{i=1}^G |o^i|} \sum_{i=1}^G \sum_{t=1}^{|o^i|} \mathbb{I}\left(H_t^i \geq \tau_\rho^{\mathcal{B}}\right)\right.$$
$$\left. \cdot \min\left(r_t^i(\theta)\hat{A}_t^i, \text{clip}\left(r_t^i(\theta), 1 - \epsilon_l, 1 + \epsilon_h\right)\hat{A}_t^i\right)\right] \tag{2}$$

## 3.3. Reinforced Training Strategy

We adopt a two-stage reinforcement training strategy with Gaussian rewards. The first stage trains a stable baseline with smooth regression-aligned feedback, while later training suffers from entropy collapse and vanishing advantages. To address this, the second stage introduces high-entropy gating and STD-guided sample filtering: the former applies policy gradients only to high-entropy tokens, and the latter removes low-variance response groups with weak gradient signals. Together, these refinements maintain gradient informativeness and improve convergence and generalization.

*Table 1.* PLCC / SRCC comparison on the technical quality assessment tasks with SOTA methods.

| Category | Methods | KonIQ | SPAQ | LIVEC | KADID | PIPAL | AGIQA3k | AVG. |
|---|---|---|---|---|---|---|---|---|
| Training-Free *Handcrafted* | NIQE (Mittal et al., 2012b) | 0.533/0.530 | 0.679/0.664 | 0.493/0.449 | 0.468/0.405 | 0.195/0.161 | 0.560/0.533 | 0.488/0.457 |
| | BRISQUE (Mittal et al., 2012a) | 0.225/0.226 | 0.490/0.406 | 0.361/0.313 | 0.429/0.356 | 0.267/0.232 | 0.541/0.497 | 0.385/0.338 |
| | *Test Setting* | *In-domain* | *Out-domain* | *Out-domain* | *Out-domain* | *Out-domain* | *Out-domain* | |
| Finetune on KonIQ *Deep Learning based method* | NIMA (Talebi & Milanfar, 2018) | 0.896/0.859 | 0.838/0.856 | 0.814/0.771 | 0.532/0.535 | 0.390/0.399 | 0.715/0.654 | 0.697/0.679 |
| | MUSIQ (Ke et al., 2021) | 0.924/0.929 | 0.868/0.863 | 0.789/0.830 | 0.575/0.556 | 0.431/0.430 | 0.722/0.630 | 0.718/0.706 |
| | CLIP-IQA+ (Wang et al., 2023) | 0.909/0.895 | 0.866/0.864 | 0.832/0.805 | 0.653/0.642 | 0.427/0.419 | 0.736/0.685 | 0.737/0.718 |
| | ManIQA (Yang et al., 2022) | 0.849/0.834 | 0.768/0.758 | 0.849/0.832 | 0.499/0.465 | 0.457/0.452 | 0.723/0.636 | 0.691/0.718 |
| | *Test Setting* | *In-domain* | *Out-domain* | *Out-domain* | *Out-domain* | *Out-domain* | *Out-domain* | |
| Jointly Finetune *MLLM-based method* | Q-Align[†] (Wu et al., 2023a) | 0.936/0.934 | 0.884/0.882 | 0.869/0.860 | 0.674/0.717 | 0.443/0.420 | 0.832/0.776 | 0.774/0.764 |
| | DeQA[†] (You et al., 2025a) | 0.928/0.908 | 0.863/0.856 | 0.847/0.866 | 0.677/0.681 | 0.411/0.390 | **0.864/0.806** | 0.763/0.751 |
| | Qwen-SFT[†] (Bai et al., 2025) | 0.850/0.823 | 0.870/0.862 | 0.792/0.784 | 0.643/0.659 | 0.423/0.404 | 0.793/0.658 | 0.729/0.698 |
| | Q-Insight[†] (Li et al., 2025c) | 0.899/0.882 | 0.899/0.897 | 0.838/0.808 | 0.627/0.666 | 0.478/**0.472** | 0.817/0.775 | 0.759/0.750 |
| | Ours | **0.941/0.927** | **0.900/0.900** | **0.880/0.857** | **0.745/0.741** | **0.481**/0.456 | 0.823/0.758 | **0.795/0.773** |

## 4. Experiment

### 4.1. Experiment Setups

**Training Datasets.** We first perform SFT on a CoT dataset comprising 41,183 examples, constructed through rejective sampling from three major sources: AVA (Murray et al., 2012), KonIQ (Hosu et al., 2020), and EvalMuse (Han et al., 2024). This dataset spans a total of 15,206 unique images—4,916 from AVA, 4,889 from KonIQ, and 5,401 from EvalMuse—and includes 3,840 samples from AVA, 15,314 from KonIQ, and 12,029 from EvalMuse. Subsequently, Qwen2.5-VL-7B (Bai et al., 2025) is fine-tuned through GRPO on three reference-scored datasets: 10K samples from AVA, 10K from EvalMuse, and 7K from KonIQ.

**Evaluation Datasets.** We evaluate our model on a wide range of datasets across three task categories: i)**Technical Quality Assessment:** Real-scene datasets include KonIQ (Hosu et al., 2020) (excluding training images), SPAQ (Fang et al., 2020), and LIVEC (Ghadiyaram & Bovik, 2015). Synthetic distortion datasets include KADID-10k (Lin et al., 2019) and PIPAL (Jinjin et al., 2020). ii)**Aesthetic Quality Assessment:** We evaluate on AVA (Murray et al., 2012) (excluding training images) and TAD66K (He et al., 2022). iii) **Text-Image Alignment Evaluation:** Evaluation is conducted on EvalMuse (Han et al., 2024) (held-out subset), EvalMi-50K (Wang et al., 2025a), T2I-CompBench (Huang et al., 2023), and GenAI-Bench (Li et al., 2024a).

**Test-Time Text-Image (T2I) Optimization.** To further verify the performance of our VisualScore for the text-image (T2I) generation task, we apply VisualScore into the test-time guided optimization strategy for T2I generation. Specifically, we evaluate its effectiveness on the Geneval Benchmark (Ghosh et al., 2023), an object-centric benchmark that evaluates compositional properties such as position, count, and color. We test the lightweight T2I model SANA-1.0-1.6B (Xie et al., 2025a) under this setting.

**Implementation Details.** We initialize the model with Qwen2.5-VL-7B and conduct supervised fine-tuning (SFT) on the synthesis CoT dataset. For the second-stage reinforcement learning, we adopt a batch size of 64 and train for 4 epochs (2 epochs for Gaussian Reward only, and 2 epochs for the STD-filter and Entropy Gating strategy). All experiments are conducted using 4 NVIDIA A100 GPUs with 80GB memory. For GRPO, we use $n = 16$ sampled responses per query, and set the hyperparameters as $\beta = 0.04$, $\epsilon = 0.2$, and the Gaussian reward decay parameter $\sigma = 0.8$.

### 4.2. Result Analysis

**Technical Quality Assessment Performance.** We categorize the compared methods into three groups: handcrafted, deep-learning-based, and MLLM-based approaches. Handcrafted methods such as NIQE (Mittal et al., 2012b) and BRISQUE (Mittal et al., 2012a) rely on manually designed features and do not involve any learning process. Deep-learning-based methods (trained on KonIQ (Hosu et al., 2020)), including NIMA (Talebi & Milanfar, 2018), MUSIQ (Ke et al., 2021), CLIP-IQA++ (Wang et al., 2023), and ManIQA (Yang et al., 2022). The MLLM-based category consists of recently proposed MLLMs like Q-Align (Wu et al., 2023a), DeQA (You et al., 2025a), Qwen-SFT (Bai et al., 2025), Q-Insight (Li et al., 2025c). These MLLMs follow the same joint finetuning setting with our proposed VisualScore. In terms of experimental setting, all models are evaluated across six benchmark datasets: KonIQ, SPAQ, LIVEC, KADID, PIPAL, and AGIQA3k, with both PLCC and SRCC used as evaluation metrics. Despite being trained jointly on three heterogeneous tasks, our method achieves consistently strong performance across all datasets, as shown in Table 1. Notably, our model reaches comparable or superior average scores to the best-performing baselines, including those trained on multiple datasets.

**Text-Image Alignment Quality Assessment Performance.** Table 2 summarizes comprehensive experimental results under two evaluation settings for text-image alignment tasks.

*Table 2.* PLCC / SRCC comparison on the text-image alignment assessment tasks.

| Category | Methods | EvalMuse-40K | EvalMi-50K | GenAI-Bench | CompBench | AVG. |
|---|---|---|---|---|---|---|
| | *Test Setting* | *Large-scale Pretraining, Out-domain testing* | | | | |
| | CLIPScore (Hessel et al., 2021) | 0.299/0.293 | 0.307/0.260 | 0.203/0.168 | 0.194/0.204 | 0.251/0.231 |
| | BLIPscore (Li et al., 2022) | 0.335/0.358 | 0.347/0.290 | 0.298/0.273 | 0.394/0.397 | 0.344/0.330 |
| | ImageReward (Xu et al., 2023) | 0.465/0.458 | 0.552/0.499 | 0.379/0.340 | 0.431/0.437 | 0.457/0.434 |
| Full-Finetune | PickScore (Kirstain et al., 2023) | 0.440/0.433 | 0.469/0.461 | 0.363/0.354 | 0.095/0.111 | 0.342/0.340 |
| | HPSv2 (Wu et al., 2023b) | 0.366/0.374 | 0.533/0.553 | 0.169/0.137 | 0.276/0.284 | 0.336/0.337 |
| | VQAScore (Li et al., 2024b) | 0.488/0.484 | 0.606/0.612 | 0.518/0.553 | 0.532/0.583 | 0.536/0.558 |
| | UnifiedReward-Llavaov (Wang et al., 2025d) | 0.710/0.722 | 0.661/0.686 | 0.606/0.621 | 0.470/0.508 | 0.612/0.634 |
| | UnifiedReward-Qwen (Wang et al., 2025d) | 0.747/0.756 | 0.736/0.717 | 0.631/0.635 | 0.627/0.648 | 0.685/0.689 |
| | *Test Setting* | *In-domain* | *Out-domain* | *Out-domain* | *Out-domain* | |
| | FGA-BLIP2 (Han et al., 2024) | 0.772/0.772 | 0.692/0.675 | 0.568/0.564 | **0.623**/0.600 | 0.664/0.653 |
| | LMM4LMM (Wang et al., 2025a) | **0.796/0.785** | 0.693/0.676 | 0.636/0.652 | 0.502/0.509 | 0.657/0.656 |
| | *Test Setting* | *In-domain* | *Out-domain* | *Out-domain* | *Out-domain* | |
| | Q-Align (Wu et al., 2023a) | 0.755/0.742 | 0.680/0.694 | 0.579/0.572 | 0.565/0.546 | 0.645/0.639 |
| | QwenSFT (Bai et al., 2025) | 0.734/0.711 | 0.718/0.683 | 0.643/0.653 | 0.566/0.598 | 0.665/0.661 |
| Few-shot Joint Finetune | DEQA (You et al., 2025a) | 0.520/0.520 | 0.462/0.431 | 0.222/0.219 | 0.326/0.325 | 0.383/0.374 |
| | Q-Insight (Li et al., 2025c) | 0.647/0.688 | 0.660/0.714 | 0.641/0.672 | 0.554/0.611 | 0.626/0.671 |
| | **Ours** | 0.764/0.775 | **0.743/0.734** | **0.674/0.679** | 0.617/**0.635** | **0.700/0.706** |

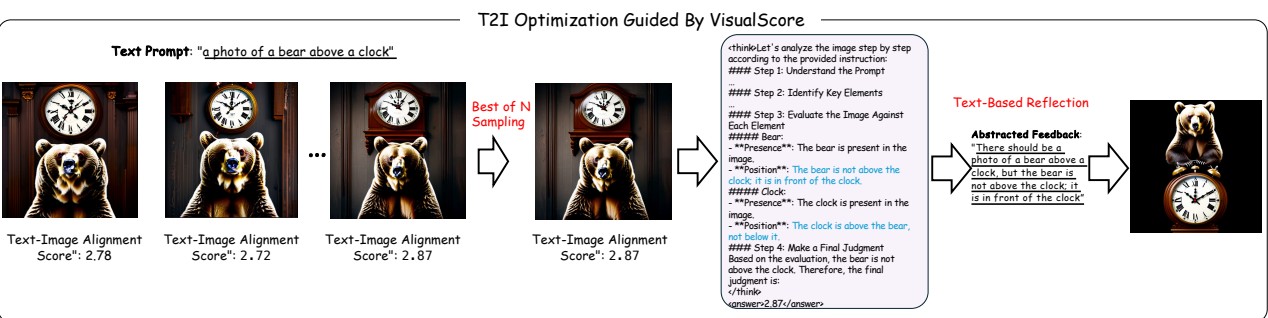

*Figure 3.* The T2I optimization process visualization guided by our proposed VisualScore.

In the *Full-finetune* category, we compare several state-of-the-art methods specifically optimized for text-image alignment. Among these, FGA-BLIP2 (Han et al., 2024) and LMM4LMM (Wang et al., 2025a) are trained on the full dataset of EvalMuse-40k (Han et al., 2024), while other approaches rely on extensive large-scale pretraining (Hessel et al., 2021; Li et al., 2023; Xu et al., 2023; Kirstain et al., 2023; Wu et al., 2023b). In contrast, methods under the *Few-shot finetune* category, including ours, are jointly trained on limited few-shot datasets, as detailed in Section 4.1. Here, EvalMuse-40K serves as the in-domain dataset, whereas EvalMi-50K, GenAI-Bench, and CompBench constitute out-of-domain benchmarks. As illustrated in Table 2, our method, despite being trained with significantly fewer data samples, achieves superior performance compared to most fully fine-tuned models on the EvalMuse-40K dataset. Additionally, our approach demonstrates excellent domain generalization capabilities, outperforming even specialized full-finetuned models and achieving state-of-the-art performance across most out-of-domain benchmarks. It is worth noting that UnifiedReward-Qwen, which shares the same backbone (Qwen2.5-VL-7B) as our model but is trained

with large-scale preference data, achieves competitive results on in-domain benchmarks. However, our method surpasses it on out-of-domain generalization (0.700 vs. 0.685 AVG PLCC), demonstrating that structured reasoning provides stronger transferability than simply scaling up preference data.

**Aesthetic Quality Assessment Performance.** Table 5 presents the evaluation of our approach on aesthetic quality assessment tasks under two distinct experimental settings: *Full-Finetune* and *Fewshot-Finetune*. As shown in Table 5, our method achieves state-of-the-art performance compared to all other methods in both in-domain and out-of-domain scenarios, clearly demonstrating the effectiveness and robust generalization capability.

**Visual Reward-Guided Test-Time T2I Scaling.** As shown in Table 4, we report the performance of T2I test-time scaling guided by our VisualScore on the GenEval (Ghosh et al., 2023) benchmark. Our method achieves the optimal performance on the GenEval benchmark through a two-stage enhancement strategy combining score-guided selection and text-guided reflection. Starting from the base

*Table 3.* Performance comparison across Stage 1 and Stage 2 reinforcement training. Stage 2 training is initialized from the corresponding Stage 1 checkpoint.

| Stage | Method | In-domain | | | Out-of-domain | | |
|---|---|---|---|---|---|---|---|
| | | AVA | EvalMuse-50k | KonIQ | TAD66k | EvalMi-50k | KADID |
| Stage 1 | GRPO+GR | **0.763 / 0.760** | **0.764 / 0.768** | 0.937 / 0.925 | **0.448/0.406** | 0.740/0.731 | 0.672/0.675 |
| | + Entro. | 0.762 / 0.762 | 0.742 / 0.746 | **0.938 / 0.925** | 0.437/0.397 | 0.716/0.715 | 0.645/0.665 |
| | + Entro. + STD | 0.757 / 0.759 | 0.748 / 0.748 | 0.934 / 0.920 | 0.436/0.398 | **0.738/0.742** | **0.735/0.731** |
| Stage 2 (from GRPO+GR) | GRPO+GR | 0.760 / 0.755 | 0.750 / 0.751 | 0.936 / 0.922 | 0.445/0.412 | 0.734/0.728 | 0.671/0.672 |
| | + Entro. | **0.767 / 0.765** | 0.767 / 0.771 | 0.940 / 0.927 | 0.450/0.409 | 0.739/0.735 | 0.735/0.740 |
| | + Entro. + STD | 0.766 / 0.763 | **0.764 / 0.775** | **0.941 / 0.927** | **0.461/0.430** | **0.743/0.734** | **0.745/0.741** |

*Table 4.* Results on the GenEval benchmark

| Generator | Overall | Single | Two | Counting | Color | Position | Attribution |
|---|---|---|---|---|---|---|---|
| SANA-1.5–4.8B (Xie et al., 2025b)[‡] | 0.76 | 0.99 | 0.95 | 0.72 | 0.82 | 0.50 | 0.54 |
| + Best-of-2048[‡] | 0.80 | 0.99 | 0.88 | 0.77 | **0.90** | 0.47 | **0.74** |
| SANA-1.0–1.6B[†] (Xie et al., 2025a) | 0.62 | 0.98 | 0.83 | 0.58 | 0.86 | 0.19 | 0.37 |
| + Best-of-20 | 0.75 | 0.99 | 0.87 | 0.73 | 0.88 | 0.54 | 0.55 |
| **+ Best-of-20 with VisualScore** | 0.78 | 0.98 | **0.96** | 0.76 | 0.86 | 0.58 | 0.60 |
| **+ Best-of-20 and Reflected 20 times with VisualScore** | **0.80** | **1.00** | **0.96** | **0.78** | 0.86 | **0.66** | 0.61 |

*Table 5.* PLCC / SRCC comparison on the aesthetic quality assessment tasks.

| Category | Methods | AVA | TAD66k | AVG. |
|---|---|---|---|---|
| | *Test Setting* | *In-domain* | *Out-domain* | |
| Full-Finetune | MUSIQ (Ke et al., 2021) | 0.738/0.726 | 0.216/0.228 | 0.477/0.477 |
| | VILA (Ke et al., 2023) | 0.664/0.658 | 0.372/0.350 | 0.518/0.504 |
| | UNIAA (Zhou et al., 2024) | 0.704/0.713 | 0.425/0.411 | 0.565/0.562 |
| | *Test Setting* | *In-domain* | *Out-domain* | |
| Few-shot Joint Finetune | Q-Align[†] (Wu et al., 2023a) | 0.747/0.729 | 0.387/0.401 | 0.567/0.565 |
| | Qwen-SFT[†] (Bai et al., 2025) | 0.650/0.629 | 0.403/0.386 | 0.526/0.507 |
| | DeQA[†] (You et al., 2025a) | 0.751/0.749 | 0.433/0.409 | 0.592/0.579 |
| | Q-Insight[†] (Li et al., 2025c) | 0.699/0.699 | 0.443/0.416 | 0.571/0.558 |
| | **Ours** | **0.766/0.763** | **0.461/0.430** | **0.612/0.597** |

*Table 6.* Comparison of reward models as rerankers on GenEval (Overall Score).

| Reward Model | Reasoning Support | Best-of-N | BoN + Revise |
|---|---|---|---|
| ImageReward (Xu et al., 2023) | × | 0.74 | – |
| UnifiedReward (Wang et al., 2025d) | ✓ | 0.77 | 0.78 |
| **VisualScore (Ours)** | ✓ | **0.78** | **0.80** |

*Table 7.* Test-time guidance on the stronger Flux.2-Klein-9B generator (GenEval Overall Score).

| Setting | GenEval Overall |
|---|---|
| Flux.2-Klein-9B (Baseline) | 0.82 |
| + Best-of-N with VisualScore | 0.88 |
| + BoN + Reflection with VisualScore | **0.89** |

SANA-1.0–1.6B model (Xie et al., 2025a), naive Best-of-20 sampling yields a solid baseline (0.75 overall). By incorporating score prediction (VisualScore) to guide the Best-of-20 sampling, we obtain a significant improvement to 0.78, outperforming the larger SANA-1.5–4.8B (0.76) despite using significantly fewer parameters and less compute. Building on this, we further apply the text-guided reflection strategy described in Reflect-DiT (Li et al., 2025b) to iteratively refine the generated outputs, achieving a new state-of-the-art score of 0.80 overall. While the larger SANA-1.5–4.8B model with Naive Best of 2048 sampling performs competitively, it is not open-sourced and requires significantly more computational resources and cost. The optimization process is shown in Fig. 3.

To further validate VisualScore as a reward model for T2I reranking, we compare it against ImageReward (Xu et al., 2023) and UnifiedReward (Wang et al., 2025d) under the same Best-of-N protocol on GenEval. As shown in Table 6, VisualScore consistently outperforms both baselines. Notably, ImageReward only outputs a scalar preference score and cannot support iterative reflection, whereas VisualScore provides structured reasoning feedback that enables the BoN+Revise strategy, achieving the highest overall score

of 0.80.

To further demonstrate the generalizability of VisualScore across different T2I generators, we evaluate it on the stronger Flux.2-Klein-9B (Labs, 2025) model. As shown in Table 7, VisualScore consistently improves generation quality on this stronger baseline, achieving +0.06 with Best-of-N and +0.07 with BoN+Reflection, confirming its effectiveness as a general-purpose reward model for T2I scaling.

**More application on test-time guidance.** To evaluate the effectiveness of test-time guidance in text-to-image (T2I) generation, we adopt a best-of-20 sampling strategy, where 20 images are generated per prompt and a scoring-based selector is used to choose the final output. We compare three selection methods on SANA 1.0: (1) Q-Align select, which uses Q-Align (Wu et al., 2023a)'s aesthetic score; (2) VisualScore-Aes select, which selects based on VisualScore's aesthetic score alone; and (3) VisualScore select, which combines both aesthetic and technical scores from VisualScore. The selected images are then evaluated using three external metrics: Q-Insight (Li et al., 2025c), DEQA (You et al., 2025a), and AesMMIT (Huang et al.,

*Table 8.* Comparison of generated images selected by different methods (20 images per prompt). Higher scores indicate better performance. And the final score is averaged on a subset of 50 prompts.

| Selection Method | Q-Insight ($\uparrow$) | DEQA ($\uparrow$) | AesMMIT ($\uparrow$) |
|---|---|---|---|
| Q-Align select | $4.165 \pm 0.198$ | $3.897 \pm 0.461$ | $0.480 \pm 0.114$ |
| VisualScore-Aes select | $4.208 \pm 0.175$ | $3.912 \pm 0.441$ | $0.513 \pm 0.116$ |
| VisualScore select | $4.264 \pm 0.175$ | $4.062 \pm 0.363$ | $0.557 \pm 0.101$ |

*Table 9.* Results on the main component of our VisualScore framework with 2 epochs training setting for stage 1 reinforcement training.

| Method | AVA | EvalMuse-50k | KonIQ |
|---|---|---|---|
| Naive GRPO | 0.733/0.732 | 0.640/0.641 | 0.899/0.882 |
| Rej. Tuning + GRPO | 0.737/0.735 | 0.689/0.706 | 0.911/0.897 |
| Rej. Tuning + GRPO with GR | **0.763/0.760** | **0.764/0.768** | **0.937/0.925** |

*Table 10.* Per-token average entropy across tasks before and after Stage 2 training with entropy gating.

| Task | After Stage 1 | After Stage 2 |
|---|---|---|
| KonIQ (Technical) | 0.0691 | 0.0687 |
| EvalMuse (Alignment) | 0.0838 | 0.0726 |
| AVA (Aesthetic) | 0.0952 | 0.0646 |

*Table 11.* Single-task vs. Multi-task training comparison (PLCC).

| Setting | KonIQ | AVA | EvalMuse |
|---|---|---|---|
| Single naive | 0.918 | 0.734 | 0.708 |
| Single + Ours | 0.936 | 0.759 | 0.731 |
| Multi naive | 0.899 | 0.733 | 0.640 |
| **Multi + Ours** | **0.941** | **0.766** | **0.764** |

2024).

As shown in Table 8, the VisualScore selector achieves the highest scores across all metrics, outperforming both Q-Align and the aesthetic-only variant. This indicates that considering both aesthetic and technical quality dimensions leads to more robust and perceptually preferred image selection. Notably, while aesthetic-only selection already improves over Q-Align, incorporating technical quality provides further performance gains, suggesting its complementary importance in high-quality T2I generation.

## 5. Ablation Study

**Single-task vs. Multi-task Training** To validate the effectiveness of multi-task joint training with our proposed stabilization mechanisms, we compare four settings in Table 11: single-task naive training, single-task with our method, multi-task naive training, and multi-task with our full framework. Multi-task naive training without stabilization performs worst due to task conflicts, while our method resolves these conflicts and achieves the best results across all benchmarks, demonstrating that unified multi-task optimization benefits from entropy gating and STD filtering.

**The effectiveness of cold-start reject sampling tuning** As shown in Table 9, cold-start reject sampling leads to consistent improvements over the naive GRPO baseline, especially on more challenging datasets like EvalMuse (0.640→0.689). These results facilitate the learning of reasoning pathway structures that reflect the underlying thought processes.

**The effectiveness of Gaussian Reward** Comparing the second and third rows of Table 9, we observe that replacing the binary reward with a Gaussian-shaped reward leads to consistent performance gains. This suggests that continuous reward modeling provides richer learning signals for the policy.

**The effectiveness of Entropy Gating and STD-guided Filtering** Table 3 reveals that directly applying entropy-based gating or STD-guided filtering during Stage 1 reinforcement learning leads to slight performance degradation across most benchmarks. And we found that these mechanisms can be counterproductive when applied to a model that has not yet acquired a strong quality assessment baseline—often resulting in unstable optimization or degraded performance. In contrast, when introduced in Stage 2 (after warming up the policy with Gaussian reward), their effects become significantly positive. The model benefits from stabilized reasoning pathways and gains better learning signals, resulting in consistent performance boosts.

To further illustrate how entropy gating balances learning across tasks, we report the per-token average entropy for each task before and after Stage 2 training in Table 10. After Stage 1, the three tasks exhibit imbalanced entropy levels (0.069 for KonIQ vs. 0.095 for AVA), indicating uneven confidence across tasks. After Stage 2 with entropy gating, the entropy values converge to a narrower range (0.065–0.073), confirming that our mechanism effectively equalizes learning signals across heterogeneous tasks.

## 6. Conclusion

In this work, we have presented VisualScore, a reward model with a structured reasoning framework for explainable and robust image quality assessment across multiple tasks. VisualScore decomposes reward modeling over key quality dimensions and leverages high-quality plan-reason trajectories for fine-grained, interpretable assessment. It combines supervised fine-tuning on structured CoT data with GRPO-based reinforcement learning for continuous score prediction, further stabilized by STD filtering and entropy gating. Extensive experiments across aesthetic, technical, and alignment evaluation tasks demonstrate that VisualScore not only achieves structured reasoning across diverse settings, but also substantially improves generalization to unseen data, suggesting a scalable path toward more robust reward models. Moreover, VisualScore can serve as a test-time reward model for guiding text-to-image (T2I) generation models, highlighting its versatility and effectiveness.

## Acknowledgements

This work was supported in part by NSFC under Grant 62371434 and U25B2010, the Fundamental Research Funds for the Central Universities (No. WK2100250064), Anhui Postdoctoral Scientific Research Program Foundation (No.2025A1015), the Postdoctoral Fellowship Program of CPSF under Grant Number GZC20252293, the China Postdoctoral Science Foundation-Anhui Joint Support Program under Grant Number 2024T017AH, and China Postdoctoral Science Foundation under Grant Number 2025M783529.

## Impact Statement

This paper presents work whose goal is to advance the field of Machine Learning. There are many potential societal consequences of our work, none which we feel must be specifically highlighted here.

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

# A. Appendix

## A.1. Use of Language Models for Manuscript Editing

This manuscript has benefited from language refinement using a large language model (LLM). The LLM was employed exclusively for linguistic improvements, such as enhancing clarity, grammar, and style, without altering the scientific content, data interpretation, or conclusions. All substantive ideas, analyses, and findings presented herein are entirely the authors' own, and the responsibility for the final content rests solely with the authors.

## A.2. Dataset Generation

The dataset generation processes for the text-image alignment task, aesthetic quality assessment task, and technical quality assessment task are illustrated in Fig. 4, Fig. 5, and Fig. 6, respectively.

Each image is paired with one or more chain-of-thought (CoT) reasoning paths generated from Qwen2.5-VL-7B. The average response lengths also vary across datasets, with AVA exhibiting longer reasoning trajectories (mean: 478.04 tokens), while KonIQ and EvalMuse maintain more concise styles (mean: 346.17 and 342.39 tokens, respectively). This diversity in both image content and reasoning style promotes broader generalization and robustness in quality assessment capabilities.

## A.3. More Experiments

**Token Entropy Visualization.** To better understand how high entropy gating impacts the model's reasoning behavior, we visualize the token-level entropy heatmaps before and after reinforcement training with entropy gating across all three tasks. As shown in Fig. 8, Fig. 9, and Fig. 10, after training, the model generates more high-entropy tokens, especially around key reasoning and decision points. This indicates increased uncertainty awareness and richer lexical exploration. Our entropy-gated training effectively mitigates entropy collapse and encourages more informative and diverse reasoning.

**Gaussian Reward Visualization.** From the Fig. 11, it can be observed that the threshold-based reward remains fixed at 1 when the error is less than 0.3, but drops abruptly to 0 once the threshold is exceeded, showing a lack of continuity. In contrast, the Gaussian-based reward decreases smoothly as the error increases, and even beyond the threshold the model still receives feedback proportional to the prediction quality. As a result, the Gaussian reward provides more informative guidance, accelerates convergence, and improves model performance on continuous-valued quality evaluation tasks.

**Hyper-Parameter.** Table 12 shows that the Gaussian reward decay achieves the best results at $\sigma = 0.8$, striking a balance between over-penalization (when $\sigma$ is too small) and under-penalization (when $\sigma$ is too large).

*Table 12.* Results on the Gaussian reward decay parameter ablation in Stage 1 Training.

| $\sigma$ | AVA | EvalMuse-50k | KonIQ |
|---|---|---|---|
| 0.6 | 0.761/0.756 | 0.734/0.756 | 0.931/0.929 |
| 0.8 | **0.763/0.760** | **0.764/0.768** | **0.937/0.925** |
| 1 | 0.741/0.735 | 0.724/0.737 | 0.913/0.912 |

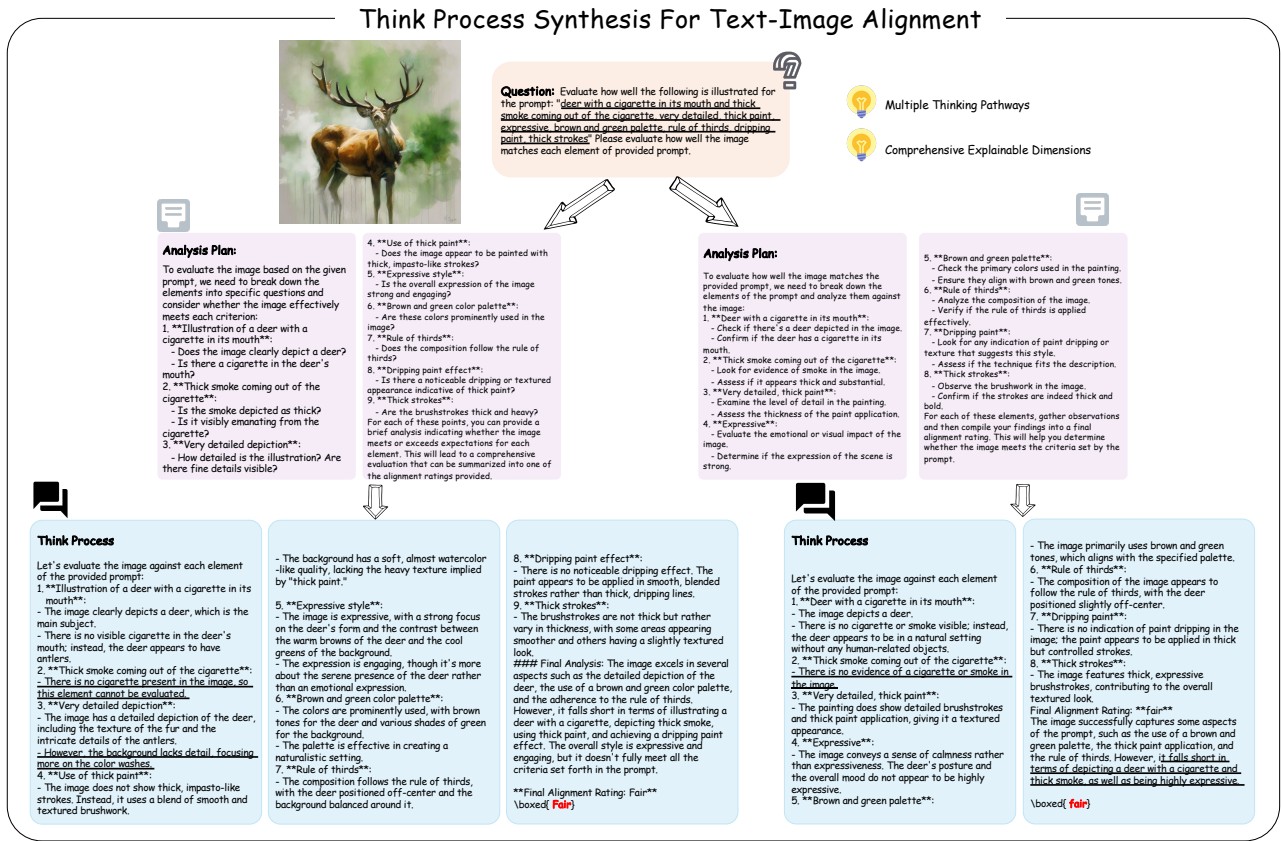

*Figure 4.* The visualization of think process synthesis for text-image alignment.

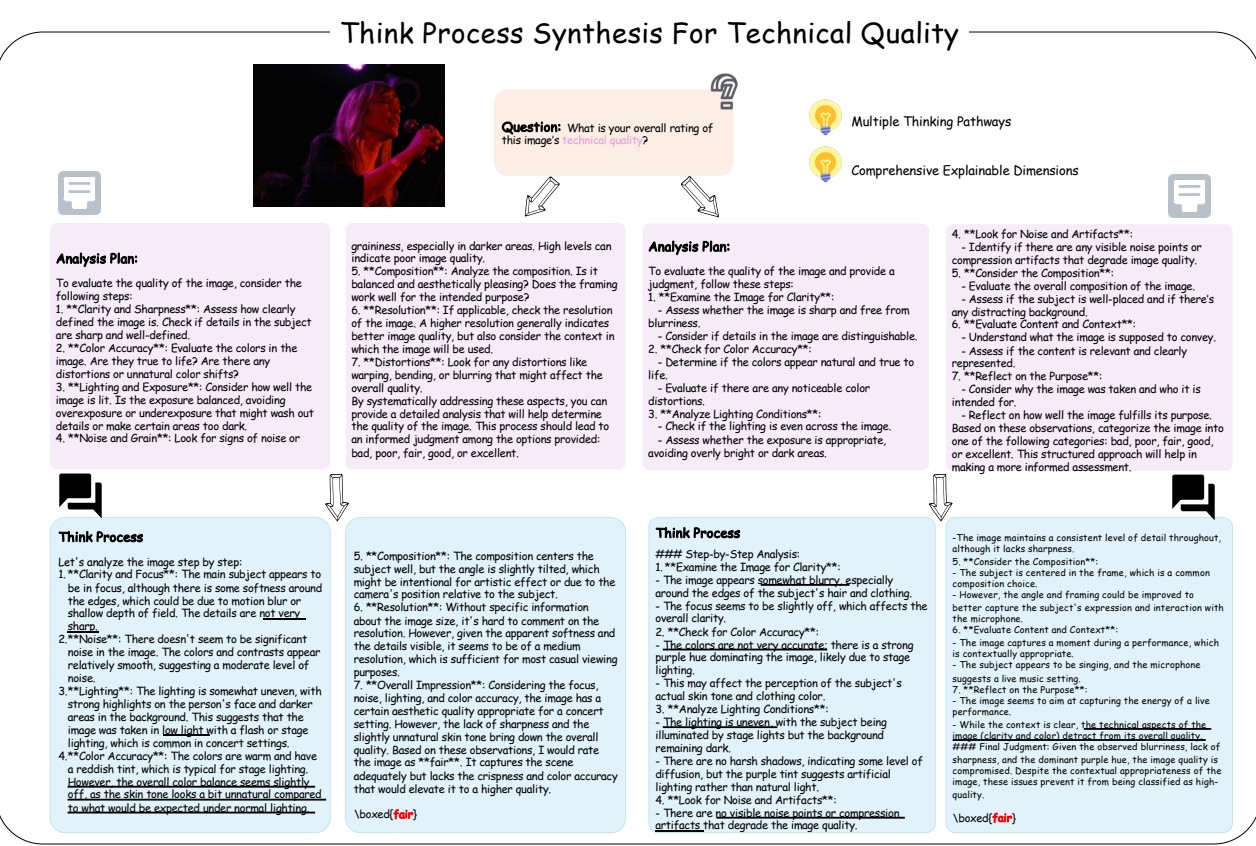

*Figure 5.* The visualization of think process synthesis for image technical quality assessment.

## Think Process Synthesis For Aesthetic Quality

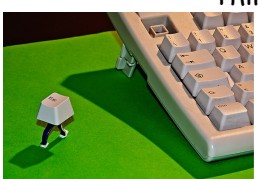

**Question:** What is your overall rating of this image's aesthetic quality? 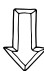

**Analysis Plan:**
Evaluating the aesthetics of an image involves several key aspects that can be systematically analyzed. Here's a structured approach for assessing the image's qualities:
### Step 1: Composition
- **Rule of Thirds**: Check if the main elements of the image align with the rule of thirds, which can create a more balanced and visually appealing composition.
- **Balance**: Assess whether the image has a sense of balance between different elements (e.g., weight distribution).
- **Framing**: Look at how the edges of the frame interact with the subjects, ensuring they don't distract from the main focus.
### Step 2: Color Harmony
- **Color Palette**: Evaluate the use of colors. Are they complementary or contrasting? Does the color scheme enhance the mood or subject matter?
- **Saturation**: Check if the colors are vivid enough but not oversaturated, which can make the image look unnatural.

- **Contrast**: Assess the contrast between different elements. Too much or too little contrast can affect the visual clarity.
### Step 3: Lighting Quality
- **Source and Direction**: Identify the light source and direction. Is it natural or artificial, and does it add depth or flatten the image?
- **Shadows and Highlights**: Evaluate the presence of shadows and highlights. Do they enhance the subject or detract from it?
- **Exposure**: Check if the exposure is appropriate. Overexposed areas may lose detail, while underexposed areas can appear dark and unclear.
### Step 4: Subject Focus
- **Depth of Field**: Assess the depth of field. Is the subject in sharp focus, or is there intentional blur (bokeh) to emphasize certain parts?

- **Clarity**: Ensure the subject is clear and free from noise or blurriness. This is especially important for macro photography or portraits.
### Step 5: Emotional Impact
- **Mood and Tone**: Consider the overall mood or tone conveyed by the image. Does it evoke a specific feeling?
- **Storytelling**: Think about whether the image tells a story or conveys a message effectively.
### Step 6: Overall Visual Appeal
- **Balance of Elements**: Assess the overall balance of all elements in the image.
- **Innovation**: Consider if image is unique or innovative in some way.
- **Impact**: How does the image make the viewer feel or react?
### Step 7: Final Evaluation
Based on the above analyses, classify the image into one of the categories: bad, poor, fair, good, or excellent. Remember, these are subjective judgments and what works for one person might not work for another.

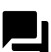

**Think Process**
### Step 1: Composition
- **Rule of Thirds**: The "ESC" key is placed slightly off-center, adhering to the rule of thirds, creating a balanced composition.
- **Balance**: The "ESC" key and the keyboard are balanced, with the keyboard taking up most of the right side of the frame, while the "ESC" key is smaller and positioned on the left.
- **Framing**: The edges of the frame do not distract from the main focus, as the green surface provides a clean background.
### Step 2: Color Harmony
- **Color Palette**: The color palette is simple, consisting mainly of white and green. The white of the keys contrasts nicely with the bright green background, creating a harmonious and visually appealing effect.
- **Saturation**: The colors are vibrant yet not oversaturated, enhancing the subject without appearing unnatural.

- **Contrast**: The contrast between the white keys and the green background is strong, making the "ESC" key stand out and the keyboard appear more prominent.
### Step 3: Lighting Quality
- **Source and Direction**: The lighting appears to be natural, possibly from a window, as indicated by the soft shadows. The direction seems to be coming from above, creating a subtle gradient on the surface.
- **Shadows and Highlights**: There are soft shadows beneath the "ESC" key and the keyboard, adding depth and dimension to the image. The highlights on the keys suggest a well-lit environment.
- **Exposure**: The exposure is appropriate, with no areas overexposed or underexposed, maintaining detail throughout the image.
### Step 4: Subject Focus
- **Depth of Field**: The "ESC" key is in sharp focus, while the rest of the keyboard is slightly blurred, creating a shallow depth of field that isolates the subject.
- **Clarity**: The subject is clear and free from noise or blurriness, with fine details visible on the keys.
### Step 5: Emotional Impact

- **Mood and Tone**: The image evokes a sense of playfulness and whimsy due to the "ESC" key being anthropomorphized with legs.
- **Storytelling**: The image tells a small, lighthearted story of a key coming alive, which is both charming and imaginative.
### Step 6: Overall Visual Appeal
- **Balance of Elements**: The composition, color harmony, and lighting work together to create a visually appealing image.
- **Innovation**: The image is unique and innovative, as it combines everyday objects in an unexpected and creative way.
- **Impact**: The viewer is likely to react positively, finding the image both amusing and aesthetically pleasing.
### Final Evaluation Based on the analysis, the image is well-composed, with effective use of color and lighting, and it successfully tells a charming story. It is visually appealing and innovative, making it a good image.

\boxed{good}

*Figure 6.* The visualization of think process synthesis for image aesthetic quality assessment.

## Technical Quality Assessment

What is your overall rating of this image's technical quality? Provide a float between 0 and 5, rounded to two decimal places. A rating of 0 represents very poor quality, while 5 represents excellent quality

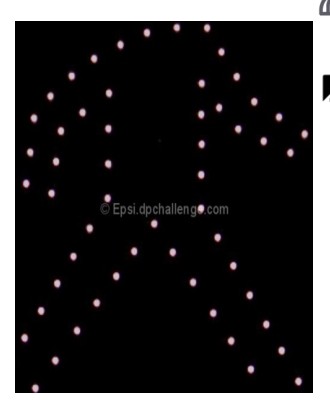

<think>Let's analyze the image step by step according to the provided instruction:

### Step 1: Understand the Criteria
- **Bad**: The image is extremely blurry, making it difficult to discern details.
- **Poor**: The image has some blurring but is still somewhat clear.
- **Fair**: The image has moderate clarity with some areas being clearer than others.
- **Good**: The image has good clarity with no noticeable issues.
- **Excellent**: The image has excellent clarity with no issues.

### Step 2: Identify Key Factors
- **Blurring**: The entire image is blurred, making it difficult to see details.
- **Compression Artifacts**: No compression artifacts are visible in the image.
- **Noise**: No noise is visible in the image.
- **Color Distortion**: No color distortion is visible in the image.
- **Other Visual Degradations**: No other issues are apparent.

### Step 3: Make a Judgment
Based on the analysis, the image is blurred and has no other visible issues. Therefore, the quality is low.

</think>
<answer>0.42</answer>

## Aesthetic Quality Assessment

What is your overall rating of this image's aesthetic quality? Provide a float between 0 and 5, rounded to two decimal places. A rating of 0 represents very poor quality, while 5 represents excellent quality

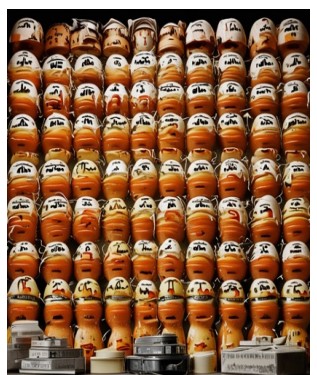

<think>Let's analyze the image step by step according to the provided instruction:

### Step 1: Understand the Question
The question asks us to evaluate the aesthetics of the image. This involves considering various aspects such as composition, color balance, lighting, and emotional impact.

### Step 2: Identify Key Elements
- **Composition**: The composition is straightforward, with a simple arrangement of dots.
- **Color Balance**: The image has a monochromatic color scheme, using only white dots against a black background.
- **Lighting**: The lighting is consistent, with no noticeable differences in brightness or shadow.
- **Emotional Impact**: The image does not convey any specific emotion; it simply presents a geometric pattern.

### Step 3: Make a Judgment
Based on the analysis, the image has a simple and straightforward composition. The color balance is consistent, and the lighting is even. However, the lack of any emotional impact and the simplicity of the pattern might make it less engaging.

### Step 4: Provide a Rating
Given the simplicity and lack of emotional impact, I would rate this image as:

</think>
<answer>1.72</answer>

## Text-Image Alignment

Does the picture accurately represent the prompt: "Alphabet soup, high quality photographic criminal evidence"? Please evaluate how well the image matches each element of provided prompt. And answer with the final alignment rating. Provide a float between 0 and 5, rounded to two decimal places. A rating of 0 represents very poor alignment, while 5 represents excellent alignment.

<think>Let's analyze the image step by step according to the provided instruction:

### Step 1: Understand the Prompt
- **Alphabet Soup**: This suggests a mix of letters and possibly numbers or symbols.
- **High Quality**: The image should be clear and detailed.
- **Photographic**: The image must be a photograph.
- **Criminal Evidence**: The image should contain elements that suggest it is related to a criminal case.

### Step 2: Identify Key Elements
- **Alphabet Soup**: The image does not contain any visible letters, numbers, or symbols.
- **High Quality**: The image is clear and detailed.
- **Photographic**: The image is a photograph.
- **Criminal Evidence**: There are no elements in the image that suggest it is related to a criminal case.

### Step 3: Evaluate the Image Against Each Element
#### Alphabet Soup:
- The image does not contain any visible letters, numbers, or symbols
#### High Quality:
- The image is clear and detailed.
#### Photographic:
- The image is a photograph.
#### Criminal Evidence:
- There are no elements in the image that suggest it is related to a criminal case.

### Step 4: Make a Final Judgment
Based on the evaluation, the image does not meet the criteria set by the prompt. It is clear and detailed, but it does not contain any visible letters, numbers, or symbols, and there are no elements that suggest it is related to a criminal case.
</think>

<answer>0.92</answer>

*Figure 7.* The visualization of our proposed VisualScore on three evaluation tasks.

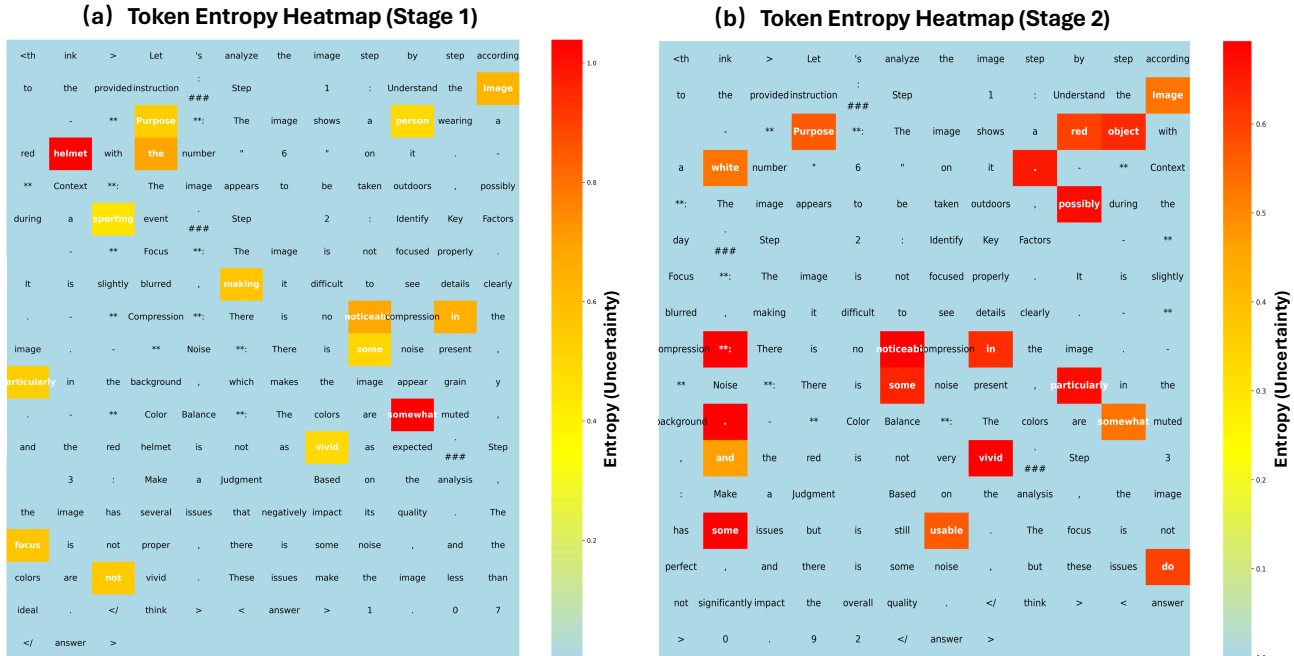

*Figure 8.* The entropy distribution for each word in the response of technical quality assessment.

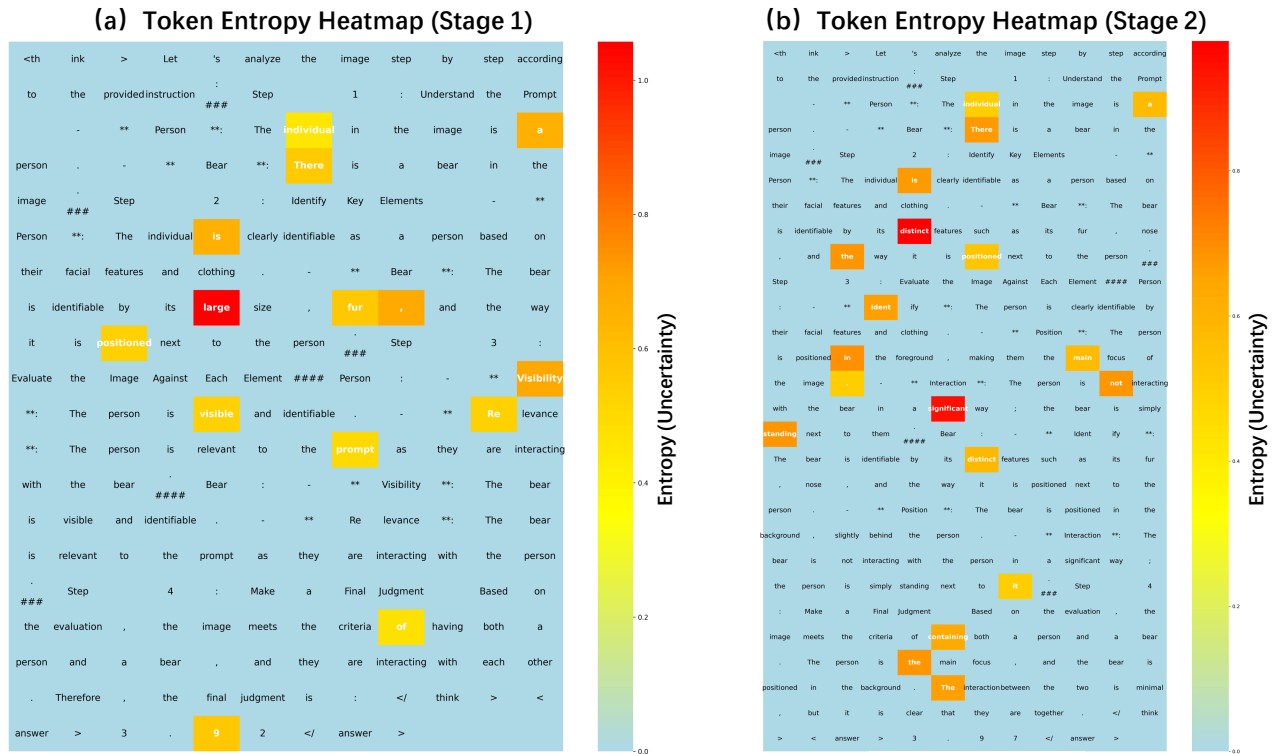

*Figure 9.* The entropy distribution for each word in the response of text-image alignment.

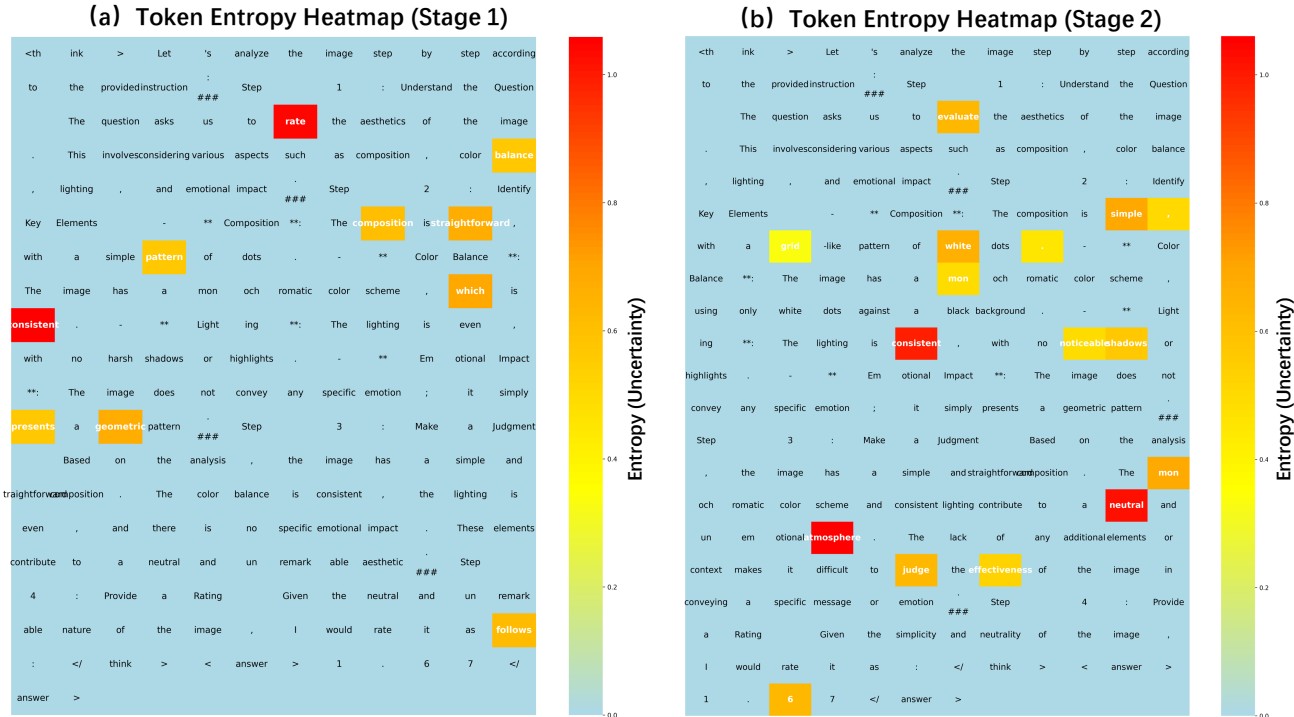

*Figure 10.* The entropy distribution for each word in the response of aesthetic quality assessment.

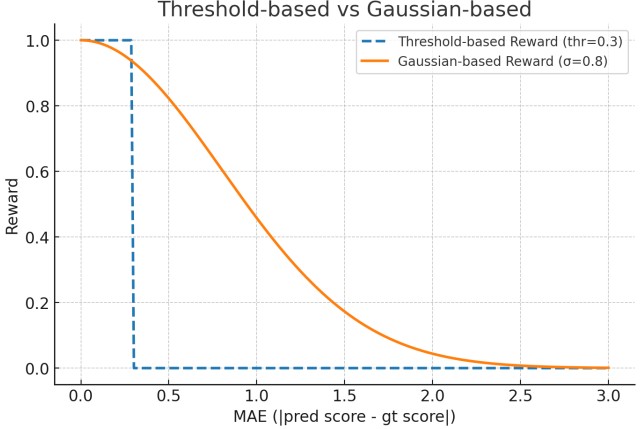

*Figure 11.* Comparison of threshold-based and Gaussian-based reward functions.

