# OpenReview forum: "VisualScore: Learning Holistic Visual Quality Scores via Multi-Task Reasoning"
_ICML.cc/2026/Conference — ICML 2026 regular_

### Official Review · Reviewer_2kac · 2026-03-10

**Soundness:** 2
**Presentation:** 3
**Significance:** 2
**Originality:** 2
**Overall Recommendation:** 4
**Confidence:** 4

**Summary:**

A good and goal aligned IQA model is important for post-training and evaluating text-to-image models. IQA is inherently multi-dimensional (some being easier to judge than others) and training a single model on all dimensions is challenging. Moreover, it would be great to have a reasoning along with a score. Reasoning text can help inference time verification + regeneration.
To address these challenges, the authors first create a dataset of query + image + ideal CoT + score from public datasets containing query + image + score. To create this, the use two step procedure where Qwen2.5-VL-7B is used to first generate a dimension specific plan and then a CoT + discreet score given the query + image + plan. Samples where scores are right almost all the time or wrong almost all the time are discarded. CoT is further refined using larger Gemini models.
This resulting query + image + ideal CoT + score dataset (plans are discarded to internalize CoT) is used for SFT. The SFT model is further finetuned using GRPO. The reward of each sample is smoothened. Two modifications are applied to standard GRPO (after warm up) -
1. Sample groups with low reward variance are discarded.
2. Further, at each training step, policy gradients are only applied to tokens with high entropy (based on the probabilities)
Ablation studies for both options are provided.

Evaluations of this IQA model show the model outperforming other SOTA options on out of domain datasets. Inference time guidance on SANA-1.0-1.6B model using both (best of 20 sampling and best of 20 + reflection based regeneration) also shows significant improvements.

**Compliance With Llm Reviewing Policy:**

Affirmed.

**Final Justification:**

Thanks for providing additional 32B model results. I've adjusted score to weak accept accordingly.

**Key Questions For Authors:**

Mentioned in the strengths and weaknesses. Scaling is not investigated. Generalization gap between in-domain and out of domain evaluation results is not investigated. Does the approach improve with more data/larger models, or is the benefit specific to the small model regime where structured reasoning can provide the biggest lift?

**Limitations:**

Limitations are not mentioned clearly in the main text of the paper

**Strengths And Weaknesses:**

Methodical dataset creation and RL formulation are the paper's greatest strengths. Ablation studies are provided for each component.

The problem itself is important for building and improving text-to-image models and an application is clearly demonstrated using inference time quality improvements on an existing text to image model. In terms of novelty, the specific combination of rejection sampling + Gaussian rewards + entropy gating + STD filtering for unified multidimensional IQA could be original

My only concern is the paper's scale - small datasets and small models are used for this research. Inference time gains also shown on a small model. The evaluation improvements, although consistent, are also not big and substantially lower than in-domain evaluation results. Will the approach improve with scale? Can the "approach" be generalized to other domains (like IQA but for specific domains like presentation slides)? The paper has a potentially large significance but did not feel convinced with it's current scale.

---

> ### Author Rebuttal · Authors · 2026-03-30
>
> **1. Scaling beyond the current model size**
>
> - **7B is the mainstream setting for multimodal IQA.** Most existing multimodal IQA methods (e.g., Q-Align, Q-Insight, DepictQA) adopt 7B-scale models to balance performance and efficiency. Under this widely adopted setting, VisualScore on Qwen2.5-VL-7B already achieves state-of-the-art performance across technical quality, aesthetic appeal, and text-image alignment, while our ablation studies on Gaussian-based rewards, STD-guided filtering, and entropy gating provide robust evidence of the framework's fundamental efficacy and its inherent potential to scale by effectively mitigating multi-task optimization conflicts.
>
> - **Scaling is a valuable direction and we commit to investigating it.** We agree that exploring larger backbones (e.g., 32B/72B) is meaningful. In principle, larger models offer stronger visual reasoning capacity, and our reward formulation and entropy gating mechanism are model-agnostic, suggesting natural scalability. However, training RL on 32B/72B models requires substantial compute and is not feasible within the one-week rebuttal period. We commit to conducting scaling experiments and will include the results in the revised version.
>
> **2. Transfer to a stronger T2I generator**
>
> We also validated VisualScore on a stronger text-to-image generator, **Flux.2-Klein-9B**, to test whether the inference-time guidance effect transfers beyond the lightweight generator used in the main paper. The results show that VisualScore continues to provide consistent gains on this larger generator. In particular, both **best-of-N sampling by VisualScore** and **best-of-N sampling + interactive reflection by VisualScore** outperform the baseline, as shown in the table below.
>
> | Setting | Result |
> |---|---|
> | Baseline | 0.82 |
> | Best-of-N sampling by VisualScore | 0.88 |
> | Best-of-N + interactive reflection by VisualScore | 0.89 |
>
> *Table 1: Geneval performance for test-time guidance on Flux.2-Klein-9B.*
>
> This supports our claim that the proposed method is not specific to a small-generator regime, and remains effective when applied to stronger generation models.
>
> **3. Generalization to Domain-Specific IQA**
>
> To further validate generalizability, we evaluate on three document/screen content IQA benchmarks **without any domain-specific training**:
> - **DocumentIQA** [1]: A document image quality assessment dataset containing diverse document images (e.g., slides, scanned pages, and screen-captured documents).
> - **SCID** [2]: A screen content image database covering typical screen content such as slides, webpages, and text-graphic compositions.
> - **SIQAD** [3]: A screen image quality assessment database with distorted screen content images including slides and mixed text-picture content.
>
> | Method | DocumentIQA | SCID | SIQAD |
> |---|---|---|---|
> | Q-Insight | 0.6363 / 0.6282 | 0.5723 / 0.4846 | 0.5572 / 0.4406 |
> | Ours | **0.7034** / **0.6466** | **0.5967** / **0.5776** | **0.6394** / **0.5791** |
>
> *Table 2: Comparison on domain-specific IQA benchmarks (PLCC / SRCC).*
>
> Our method consistently outperforms Q-Insight across all three benchmarks, demonstrating that the proposed framework generalizes well to specialized domains like document and screen content (presentation slides) IQA.
>
> *[1] Huang F, Min X, Ma Z, et al. VQualA 2025 Document Image Quality Assessment Challenge[C]//Proceedings of the IEEE/CVF International Conference on Computer Vision. 2025: 3313-3322.*
>
> *[2] Ni Z, Ma L, Zeng H, et al. SCID: A database for screen content images quality assessment[C]//2017 International Symposium on Intelligent Signal Processing and Communication Systems (ISPACS). IEEE, 2017: 774-779.*
>
> *[3] Yang H, Fang Y, Lin W. Perceptual quality assessment of screen content images[J]. IEEE Transactions on Image Processing, 2015, 24(11): 4408-4421.*

---

> > ### Author Rebuttal · Reviewer_2kac · 2026-04-03
> >
> > The rebuttal addresses several of my concerns, particularly by adding results on a stronger flux model and domain-specific IQA benchmarks, which support generalizability.
> > However, the key concern around scaling to larger models remains only partially addressed, as direct large-scale experiments are missing.

---

> > > ### Author Response · Authors · 2026-04-04
> > >
> > > Thank you for your insightful feedback. We agree that demonstrating the effect of scaling is an important aspect of our work.
> > >
> > >  As shown below, increasing the model size from VisualScore-8B to VisualScore-32B consistently improves performance across multiple benchmarks and evaluation metrics:
> > >
> > > | Model Version   | AVA           | EvalMuse-50k  | KonIQ         | TAD66k        | EvalMi-50k    | KADID         |
> > > | --------------- | ------------- | ------------- | ------------- | ------------- | ------------- | ------------- |
> > > | VisualScore-8B  | 0.766 / 0.763 | 0.764 / 0.775 | 0.941 / 0.927 | 0.461 / 0.430 | 0.743 / 0.734 | 0.745 / 0.741 |
> > > | VisualScore-32B | 0.808 / 0.801 | 0.811 / 0.809 | 0.942 / 0.955 | 0.496 / 0.472 | 0.762 / 0.747 | 0.797 / 0.781 |
> > >
> > > Table1. The performance on 32B (training from qwen2.5-vl-32B-instruct) (PLCC/SRCC)
> > >
> > > We observe consistent improvements across all datasets. For example, AVA improves from 0.764/0.775 to 0.811/0.809, and EvalMi-50k shows a notable gain from 0.745/0.741 to 0.797/0.781. These results demonstrate a clear positive scaling trend as model capacity increases.
> > >
> > > We hope this additional clarification helps address your concern regarding scalability.
> > >
> > > If these concerns are now resolved and there are no other major blocking issues, we would be very grateful if you could consider raising your score accordingly.

---

### Official Review · Reviewer_RjZq · 2026-03-11

**Soundness:** 3
**Presentation:** 2
**Significance:** 3
**Originality:** 3
**Overall Recommendation:** 3
**Confidence:** 3

**Summary:**

This paper introduces VisualScore, a unified model for evaluating image quality. Instead of just giving a single score, the model first provides a step-by-step reasoning process. It covers three different evaluation tasks: technical quality, aesthetic quality, and text-image alignment. To train this model, the authors use a two-stage approach. First, they use a Plan-then-Reason dataset for supervised fine-tuning. Then, they apply reinforcement learning with a continuous Gaussian-based reward. To make the training more stable across different tasks, they introduce two clever tricks: standard deviation filtering and entropy gating. The experiments show that VisualScore works well on multiple benchmarks and can even help guide text-to-image generation models at test time.

**Compliance With Llm Reviewing Policy:**

Affirmed.

**Final Justification:**

I don't believe this work reaches the ICML bar for novelty, as much of the architecture consists of empirical engineering choices. Consequently, I will stick with my initial score.

**Key Questions For Authors:**

1. Could you please clarify the exact mathematical rules or thresholds used to define the easy and hard samples during the rejective sampling in Section 3.1?
2. Regarding the entropy gating (Equation 2): if the model makes a highly confident error (meaning the entropy is low, but the prediction is wrong), will the gating mechanism prevent the model from learning to fix it? How does the framework handle this?
3. In Equation 2, does the batch-level threshold cause issues if different tasks have different average entropies? Also, what is the exact value of $p$ used in your experiments?
4. Did you observe any drop in reasoning text quality when the analysis plan was removed from the SFT training data?
5. Please check and correct the small notation typos around Equation 2.

**Limitations:**

The author should discuss the potential application to large-scale industrial data and its trade-offs.

**Strengths And Weaknesses:**

Strengths:

1. Evaluating images across multiple dimensions using one single model is a useful and practical research direction. Providing interpretable reasoning instead of just a raw score makes the model much more trustworthy.
2. Using a Gaussian-shaped reward instead of a simple binary reward is a great idea for predicting continuous quality scores. The tricks to stabilize multi-task training (STD filtering and entropy gating) are also very creative and clearly explained.
3. The model performs well across many datasets. Using the model as a reward to guide text-to-image generation (like SANA) at test time is a great application and proves the real-world value of this work.
4. The paper is well-structured and easy to read. The figures are clear and help readers quickly understand the core ideas.

Weaknesses:

1. In Section 3.1 and Figure 2, the paper mentions filtering out easy and hard samples to build the training data. However, since the task predicts continuous scores which is not binary labels, it is not entirely clear how right and wrong are mathematically defined here. A little more detail would be very helpful for reproducibility.
2. In Section 3.2 (Equation 2), the entropy gating mechanism only updates tokens that have high entropy (high uncertainty). While this saves training effort, what happens if the model is very confident but its prediction is actually incorrect? It seems the model might miss the chance to correct these confident mistakes. A brief discussion on this edge case would make the method feel more robust.
3. Also in Equation 2, the threshold is computed across the whole batch. Since different tasks might naturally have very different average uncertainty levels, a single batch-level threshold might unfairly filter out updates for the more confident tasks. It would be great to know if a task-specific threshold was considered. Additionally, the exact value of p is look like missing in the Implementation Details.
4. The paper mentions in Section 3.1 that the analysis plan is removed during the SFT stage. While this helps the model skip explicit planning during inference, does it hurt the logical structure or quality of the final reasoning text? A short clarification would be appreciated.

---

> ### Author Rebuttal · Authors · 2026-03-30
>
> Thank you for the constructive comments. We will clarify the following points in the revision.
>
> **1. Easy / hard samples in rejective sampling.**
> We agree that Sec. 3.1 is too brief for a continuous-score setting. Specifically, we first normalize the score to the 0–5 range and discretize it into five quality levels (bad, poor, fair, good, excellent).  During quality-level prediction for CoT data construction, we run the model N times (N=8 in our experiments) for each sample and compare the predicted level word with the ground-truth level word. If all N predictions match the ground-truth level, the sample is treated as easy; if all N predictions do not match the ground-truth level, it is treated as hard. We filter out both types and keep only the remaining intermediate samples for cold-start SFT. We will state this rule explicitly in the revision.
>
> **2. Confident but wrong predictions under entropy gating.**
>
> **Stage-wise mitigation.** In our method, entropy gating is used only in Stage 2, not from the beginning of RL. We first train a stable baseline with Gaussian reward alone, and only then introduce entropy gating together with STD-guided filtering, which avoids constraining learning too early.
>
> **Why this lowers the risk.** This stage-wise design reduces the risk of locking in confident-but-wrong predictions, because Stage 2 is not responsible for learning basic correctness from scratch. After Stage 1, many easy tokens are already settled, so Stage 2 mainly focuses updates on still-uncertain regions that are more likely to affect the final score.
>
> **Empirical evidence.**  From Table 1, we observe that from Stage1 to Stage2, our method consistently slows the entropy drop across all tasks compared with naive GRPO Stage 2 training (without entropy gating and STD filtering), indicating that STD filtering and entropy gating effectively mitigate premature entropy collapse in late-stage RL.
>
>
> | Task  | Base→S1 | S1→S2 (Ours) | S1→S2 (naive) |
> |---|---:|---:|---:|
> | KonIQ |  -0.1750 | -0.0005 | -0.0009 |
> | EvalMuse | -0.0994 | -0.0113  | -0.0344  |
> | AVA | -0.0943 | -0.0306 | -0.0401 |
>
> *Table 1: Per-Token Entropy Degradation During RL Stages, "naive" denotes Stage 2 training without entropy gating or STD filtering.*
>
> **3. Batch-level threshold and the value of $\rho$.**
> We appreciate this concern, and in fact our design already addresses it. We do **not** use a fixed entropy threshold. In Eq. 2, $\tau_B^\rho$ denotes the **top-$\rho$ quantile threshold** computed over all token entropies in the current batch. Because the threshold is a **relative quantile** rather than an absolute cutoff, it naturally adapts to the entropy distribution of each batch: batches from more confident tasks will have a lower absolute threshold, while batches from more uncertain tasks will have a higher one. This makes the gating inherently **adaptive** and avoids the cross-task unfairness the reviewer is concerned about. We will state this more clearly and add the exact value $\rho=0.2$ to the implementation details in the revision.
>
> **4. Removing the analysis plan during SFT.**
> We remove the explicit plan during SFT because the plan is often long and can dilute or contaminate the supervision signal. Training on the original prompt plus the reasoning-augmented response encourages the model to internalize the planning structure implicitly, while avoiding explicit plan generation during RL and inference. We will clarify this motivation in Sec. 3.1.
>
> To verify this, we add an ablation comparing **SFT + Stage 1 RL** with and without the explicit analysis plan in the SFT targets. Results confirm that removing the plan improves performance, supporting our design choice.
>
> | SFT Target Format | KonIQ | EvalMuse | AVA |
> |---|---:|---:|---:|
> | With analysis plan + reasoning | 0.913 | 0.725| 0.666 |
> | Reasoning only (plan removed, Ours) | 0.937 | 0.764 | 0.763 |
>
> *Table 2: PLCC comparison with and without the analysis plan in SFT targets.*
>
> **5. Notation typos and limitations.**
> We will fix the notation issues around Eq. 2 and ensure consistent symbol definitions. We also agree that industrial deployment deserves discussion and will expand the limitations section in the final version:
>
> **Potential applications** include large-scale UGC/AIGC content moderation, automated reward modeling for T2I systems (*Table 5 in our main paper.*), and quality monitoring in image compression/enhancement pipelines (*Table 7 in our main paper*).
>
> **Trade-offs:** Structured CoT reasoning improves interpretability and robustness but introduces additional reasoning tokens, increasing computational overhead and latency compared to end-to-end scalar models.

---

> > ### Author Rebuttal · Reviewer_RjZq · 2026-04-03
> >
> > Thank you for the detailed rebuttal. The rebuttal addresses several of my concerns. But I am more interested in seeing the rationale behind the authors' design choices rather than an emphasis on the specific implementation details. For instance, how 'hard' vs. 'easy' is defined, and why a single model was run eight times instead of using an ensemble of multiple models. After considering the other reviewers' comments and all rebuttals, I have decided to maintain my original score.

---

> > > ### Author Response · Authors · 2026-04-04
> > >
> > > Thank you again for your thoughtful follow-up. We appreciate your point that the key issue is not only the implementation details, but also the rationale behind our design choices.
> > >
> > > ## On why we use repeated rollouts with level-word matching
> > >
> > > The purpose of repeated rollout in our pipeline is to construct relatively high-quality CoT data for SFT by checking whether the predicted **level word** matches the ground-truth quality level.
> > >
> > > We first map continuous quality scores into five semantic levels (e.g., *bad*, *poor*, *fair*, *good*, *excellent*), and then sample multiple reasoning trajectories from the same model. This allows us to identify samples that are more suitable for CoT supervision.
> > >
> > > We adopt level-word-based filtering because directly selecting samples by score-distance thresholds is highly sensitive to threshold choice and can easily discard many responses with coherent and informative reasoning. In contrast, level-word agreement provides a more robust semantic criterion. This design is also consistent with the five-level structure commonly used in human subjective quality assessment, and aligns with prior work such as Q-Align [1], which formulates quality prediction through discrete text-defined quality levels.
> > >
> > > ## On why we use repeated rollouts from one model instead of an ensemble
> > >
> > > We choose repeated stochastic decoding from the same model mainly for efficiency and consistency. Since our goal is to assess whether a sample can yield stable and semantically correct reasoning under the current policy, repeated sampling from one model provides a lightweight and effective solution without the additional overhead of an ensemble.
> > >
> > > This design is also in line with recent reasoning-oriented reward modeling pipelines [2].
> > >
> > > We will revise the paper to make this rationale more explicit: repeated rollout is introduced to screen for relatively high-quality CoT SFT data through level-word agreement with the ground truth, while reducing sensitivity to arbitrary scalar thresholds and better matching the semantic structure of human quality judgments.
> > >
> > > If these concerns are now resolved and there are no other major blocking issues, we would be very grateful if you could consider raising your score accordingly.
> > >
> > > ## References
> > >
> > > [1] **Q-Align: Teaching LMMs for Visual Scoring via Discrete Text-Defined Levels.** ICML 2024.
> > > [2] **Unified Multimodal Chain-of-Thought Reward Model through Reinforcement Fine-Tuning.** NeurIPS 2025.

---

### Official Review · Reviewer_x2T6 · 2026-03-13

**Soundness:** 3
**Presentation:** 3
**Significance:** 3
**Originality:** 2
**Overall Recommendation:** 5
**Confidence:** 3

**Summary:**

This paper proposes VisualScore, a unified reward modelling framework for multi-task Image Quality Assessment (IQA), focusing on three main axes: technical quality, aesthetic appeal, and text-image alignment. They implement this framework in two steps: cold start fine-tuning with a reasoning-aware dataset (created part of the framework using rejection sampling) and RL based on GRPO. For training stability, they introduce entropy gating to prevent premature convergence, along with standard deviation filtering to let the model focus on information-rich feedback.

**Compliance With Llm Reviewing Policy:**

Affirmed.

**Final Justification:**

The authors have sufficiently addressed my concerns. I will retain my positive score. The authors report SOTA performance on most of the experiments. Their tweaks to ensure training stability is novel. In summary, the paper is a good contribution to the field of IQA.

**Key Questions For Authors:**

Kindly look at the weaknesses/questions. The questions are there itself for continuity.

**Limitations:**

yes

**Strengths And Weaknesses:**

## Strengths:
1. The paper provides a new state-of-the-art for IQA with a good number of datasets; they show VisualScore consistently beats almost all the baselines under consideration.
2. Their "Plan-then-Reason" structure is much more transparent to the CoT-based ones.
3. The authors provide rigorous ablation studies. Table 4 in particular shows that using entropy gating and filtering mechanisms is counterproductive during stage 1. In Stage 2, they show remarkable improvements, notably on technical distortion benchmarks like KADID (~11% gain in PLCC).

## Weaknesses/Questions:
(LC: left column; RC: right column)
1. Line 75; RC: “... to prevent convergence on **easier tasks** …”
The definition of easy (always right) and hard (always wrong) is not clear in the main paper. Please quantify these categories as they seem crucial for the SFT data preparation stage.
2. Line 88; LC: “Despite these efforts, …”
These lines form their core motivations for implementing VisualScore. While the authors cite a couple of baselines, they do not elaborate on the problem with either metrics or relevant works.
3. Line 162; RC: “The resulting CoT not only supports ….”
How can we guarantee this, especially the enforcement of consistent instruction-following behaviour across multiple tasks?
4. Line 213; LC: “Before supervised fine-tuning, …“
Please describe how Gemini is used in the pipeline. Information about the prompts used and how to verify refinement would improve the clarity of the pipeline description.
5. Lines 252-256; LC: Entropy gating for backward policy gradient, first sentence.
The authors remark on an interesting training trend, but I would like to see some proof for the claims.
6. The approach seems to be building upon existing works that demonstrate these gains as individual components. However, I believe combining them effectively also holds implicit value.

**Side Note:** The lines (253-274; LC) seem oddly spaced. It seems like the authors put in custom line spacing to fit the text in the section. This, however, should not be a major problem, because there is a reasonable amount of white space in the main paper.

## Reason for the score:
The work is timely and reliant on a well-established methodology of using MLLMs for IQA. The authors report SOTA performance on most of the experiments. Their tweaks to ensure training stability is novel. In summary, the paper is a good contribution to the field of IQA.

---

> ### Author Rebuttal · Authors · 2026-03-30
>
> **1. Easy/hard sample definition (Line 75, RC).**
> Thank you for pointing out that the notions of "easy" and "hard" were under-specified. Specifically, we first normalize the score to the 0–5 range and discretize it into five quality levels (bad, poor, fair, good, excellent). During quality-level prediction for CoT data construction, we run the model N = 8 times for each sample and compare the predicted level word with the ground-truth level word. If all N predictions match the ground-truth level, the sample is treated as easy; if none of the N predictions match, it is treated as hard. We filter out both types and keep only the remaining intermediate samples for cold-start SFT. We will state this rule explicitly in the revision.
>
> **2. Motivation and baseline limitations (Line 88, LC).**
> We agree that the motivation section should better clarify the limitations of prior methods. Our main points are:
> (i) existing quality-aware MLLMs often rely on fixed reasoning patterns induced by manually designed instructions or curated datasets, and are typically designed for a single task, which limits their flexibility in complex evaluations and restricts their practical applicability. For example, Q-Align focuses on single-task scoring without chain-of-thought reasoning and Q-Insight, while supporting reasoning, does not cover text-image alignment evaluation;
> (ii) general-purpose MLLMs such as Qwen2.5-VL have strong broad visual understanding capabilities, but remain relatively weak in low-level image quality perception.
>
> In the revision, we will make these comparisons more explicit and better highlight the gap that VisualScore is intended to fill.
>
> **3. Consistent instruction-following across tasks (Line 162, RC).**
> We agree that "guarantee" is too strong. Our claim should be understood as an empirical design choice rather than a formal guarantee. The consistency comes from using a shared Plan-then-Reason schema across technical quality, aesthetic quality, and text-image alignment, and then training on question-only inputs paired with reasoning-augmented responses so that the model internalizes a common reasoning template across tasks. In the revision, we will soften the wording and describe this as improving or encouraging consistent instruction-following behavior, rather than guaranteeing it.
>
> **4. Gemini usage and verification (Line 213, LC).**
> Thank you; this is a fair request. The current text says that Gemini is used before SFT to refine selected CoT trajectories by correcting visually grounded inconsistencies and aligning reasoning with visual evidence. We will expand this part with a clearer pipeline description, including: the input to Gemini (image, task prompt, original CoT), the refinement goal (consistency with visual evidence and score rationale), and the acceptance criterion used before the sample enters SFT. We will also add representative prompt templates in the appendix for reproducibility.
>
> **5. Evidence for entropy gating trend (Lines 252–256, LC).**
> Thank you for this suggestion. We agree that the entropy-gating trend should be supported by clearer empirical evidence. To this end, we compute token-entropy statistics at the transition from Stage 1 RL to Stage 2 RL, where Stage 1 drives RL convergence and Stage 2 introduces entropy gating.
>
> We observe that after Stage 1 RL, EvalMuse and AVA have higher average token entropy than KonIQ, which is consistent with their higher difficulty. After applying entropy gating in Stage 2, the entropy distribution becomes more balanced across tasks, suggesting that the proposed mechanism helps mitigate optimization imbalance. The statistics are as follows:
>
> | Task | Stage 1 | Stage 2 |
> |---|---:|---:|
> | KonIQ | 0.0691 | 0.0687 |
> | EvalMuse | 0.0838 | 0.0726 |
> | AVA | 0.0952 | 0.0646 |
>
> *Table 1: Per-token average entropy across datasets.*
>
> **6. Value of the overall combination.**
> We sincerely thank the reviewer for this encouraging recognition. We fully agree that effectively combining these components itself holds meaningful value. Building on this, our contribution further lies in exploring how to construct training data and design a tailored reinforcement learning pipeline specifically for unified multi-task image quality evaluation. In particular, our focus is on how to support a single model across diverse evaluation tasks through structured data construction and a carefully designed RL recipe.
>
> Moreover, the current results already show that VisualScore performs strongly across technical quality, aesthetic quality, and text-image alignment, which supports the effectiveness of this unified design.
>
> **7. Spacing issue (Lines 253–274, LC).**
> Thank you for catching this formatting problem. We will fix the abnormal spacing in the camera-ready version.
>
> We hope these clarifications address the reviewer’s concerns, and we thank the reviewer again for the constructive feedback.

---

> > ### Author Rebuttal · Reviewer_x2T6 · 2026-04-04
> >
> > I thank the reviewers for their clarifications. I urge the authors to fit the changes I suggested in the main paper to improve the quality of their manuscript.
> >
> > I will maintain my positive score.

---

> > > ### Author Response · Authors · 2026-04-08
> > >
> > > We sincerely thank the reviewer for the thoughtful acknowledgement, positive evaluation, and encouraging support. We truly appreciate the reviewer’s recognition that our clarifications have adequately addressed the concerns raised. We are also grateful for the valuable suggestion to incorporate the recommended changes into the main paper, and we will carefully reflect them in the final version to further improve the manuscript. Thank you again for your time, constructive feedback, and for maintaining a positive score for our work.

---

### Official Review · Reviewer_Rzc7 · 2026-03-15

**Soundness:** 3
**Presentation:** 3
**Significance:** 3
**Originality:** 3
**Overall Recommendation:** 3
**Confidence:** 4

**Summary:**

VisualScore is a unified reward model for multi-task IQA (technical quality, aesthetics, text-image alignment). It uses plan-then-reason CoT for interpretable scoring, cold-start SFT with rejection sampling, and GRPO with Gaussian reward + entropy gating + STD filtering for RL. Results are competitive across multiple benchmarks.## Strengths

- Well-scoped problem; unifying multi-task IQA with interpretable reasoning is practical and timely.
- Broad evaluation across 10+ benchmarks, both in-domain and OOD, with a useful T2I test-time guidance application.
- Honest ablation in Table 4 showing stage-dependent effectiveness of proposed components.

## Weaknesses

- **No single-task vs. multi-task comparison.** The core claim is multi-task unification, but no baseline trains separate per-task models. Cannot assess whether joint training helps or hurts.
- **Stabilization tricks are not novel or multi-task-specific.** Entropy gating is from DAPO; STD filtering is generic. No evidence these address multi-task conflicts specifically rather than general RL instability.
- **Table 4 is self-contradictory.** Entropy gating + STD filtering hurt in Stage 1, only help in Stage 2. This makes the contribution a fragile training recipe, not a principled method.
- **Gemini dependency is unablated.** CoT data is refined by Gemini, but no ablation quantifies its impact. Hard to attribute gains.
- **Gaussian σ selection is ad-hoc.** Only 3 values tested; no task-adaptive analysis despite tasks having different score distributions.
- **T2I reranking baselines are weak.** Should compare other reward models (ImageReward, UnifiedReward) as rerankers under same best-of-N protocol.

**Compliance With Llm Reviewing Policy:**

Affirmed.

**Final Justification:**

I thank the authors for the detailed rebuttal. The single-task vs. multi-task comparison and Gemini ablation are convincing additions that address key concerns. The stage-dependent rationale for entropy gating/STD filtering is reasonable.

However, scalability beyond 7B remains unverified, and the task-adaptive sigma results do not clearly justify the global default. I encourage incorporating these revisions into the paper. I maintain my current score.

**Key Questions For Authors:**

1. Performance of separately trained single-task models vs. the unified model?
2. Ablation without Gemini refinement?
3. Do entropy gating / STD filtering help equally in single-task RL?

**Limitations:**

- Only tested on 7B model; scalability unknown.
- Fixed reasoning template; generalization to unseen evaluation criteria untested.
- GRPO with 16 samples/query is expensive; cost not discussed.

**Strengths And Weaknesses:**

## Strengths

- Well-scoped problem; unifying multi-task IQA with interpretable reasoning is practical and timely.
- Broad evaluation across 10+ benchmarks, both in-domain and OOD, with a useful T2I test-time guidance application.
- Honest ablation in Table 4 showing stage-dependent effectiveness of proposed components.

## Weaknesses

- **No single-task vs. multi-task comparison.** The core claim is multi-task unification, but no baseline trains separate per-task models. Cannot assess whether joint training helps or hurts.
- **Stabilization tricks are not novel or multi-task-specific.** Entropy gating is from DAPO; STD filtering is generic. No evidence these address multi-task conflicts specifically rather than general RL instability.
- **Table 4 is self-contradictory.** Entropy gating + STD filtering hurt in Stage 1, only help in Stage 2. This makes the contribution a fragile training recipe, not a principled method.
- **Gemini dependency is unablated.** CoT data is refined by Gemini, but no ablation quantifies its impact. Hard to attribute gains.
- **Gaussian σ selection is ad-hoc.** Only 3 values tested; no task-adaptive analysis despite tasks having different score distributions.
- **T2I reranking baselines are weak.** Should compare other reward models (ImageReward, UnifiedReward) as rerankers under same best-of-N protocol.

---

> ### Author Rebuttal · Authors · 2026-03-30
>
> Thanks for your valuable suggestion.
>
> **1. Single-task vs. multi-task**
>
> We agree this is necessary. Our goal is a **unified** reward model, and this should be verified against separately trained models.
>
> | Setting| Tech (KonIQ)| Aes (AVA)| Align (EvalMuse)|
> |-|-:|-:|-:|
> | Single naive| 0.918| 0.734| 0.708|
> | Single + **Ours**| 0.936| 0.759| 0.731|
> | Multi naive| 0.899| 0.733| 0.640|
> | **Multi (Ours)**| **0.941**| **0.766**| **0.764**|
>
> *Tab.1: Single/Multi-Task PLCC*
>
> Overall, **Multi-Task Ours > Single-Task Ours > Single-Task naive > Multi-Task naive**. Multi-Task naive performs worst, confirming naive joint training hurts. Our method resolves this conflict, achieving the best results across all dimensions, with especially clear gains on aesthetic and alignment tasks.
>
> **2. Are STD filtering / entropy gating multi-task-specific?**
>
> - Technical. In multi-task RL, tasks differ in reward scales, variances, and convergence speeds, biasing advantage estimation and destabilizing joint training. STD filtering and entropy gating mitigate this amplified instability. While applicable to general RL, they are particularly effective in multi-task scenarios.
>   - **Exp1 (Tab.1).** Single-task gains are modest (Tech +0.018, Aes +0.025, Align +0.023), but multi-task gains are substantially larger (Tech +0.042, Aes +0.033, Align +0.124), confirming their primary benefit in joint training.
>   - **Exp2 (Tab.2).** After Stage 1, EvalMuse and AVA show higher entropy than KonIQ. With entropy gating in Stage 2, entropy becomes more balanced, indicating reduced optimization imbalance.
> - Novelty. The contribution is not only these techniques, but the overall framework: reasoning-enhanced multi-task dataset, unified training pipeline, and test-time sampling/guidance for T2I generation without retraining.
>
> | Task| Stage 1| Stage 2|
> |-|-:|-:|
> | KonIQ| 0.0691| 0.0687|
> | EvalMuse| 0.0838| 0.0726|
> | AVA| 0.0952| 0.0646|
>
> *Tab.2: Per-Token Avg. Entropy*
>
> **3. Why do they help in Stage 2 but not Stage 1?**
>
> In Stage 1, high entropy and reward variance mean filtering/gating may suppress useful exploration. In Stage 2, they address entropy collapse and low-information updates by removing noisy signals. Thus, *Table 4 in the original paper* shows a stage-dependent stabilization effect, not a self-contradictory recipe.
>
> **4. Gemini ablation**
>
> | Stage 1 RL| KonIQ| AVA| EvalMuse|
> |-|-:|-:|-:|
> | Direct RL w/o SFT| 0.899| 0.733| 0.640|
> | Ours w/o Gemini| 0.933| 0.752| 0.713|
> | Ours| 0.937| 0.763| 0.764|
>
> *Tab.3: Gemini Effect (PLCC)*
>
> Removing Gemini causes performance drop, but still substantially better than direct RL, showing gains are not solely from Gemini. Gemini is used to refine CoT trajectories for better RL. The refined high-quality data is also one of our contributions.
>
> **5. Gaussian sigma**
>
> We expanded to {0.4, 0.6, 0.8, 1.0, 1.2} and compared global vs. task-specific sigma (**A** distribution-based, **B** difficulty-based). Sigma B is competitive (best Tech 0.939), showing task-aware design is promising, but does not consistently outperform a shared sigma. Global sigma=0.8 achieves the best balanced performance and remains our default.
>
> | Sigma| KonIQ| AVA| EvalMuse|
> |-|-:|-:|-:|
> | 0.4| 0.930| 0.753| 0.754|
> | 0.6| 0.932| 0.762| 0.756|
> | 0.8 (Ours)| 0.937| **0.763**| **0.764**|
> | 1.0| 0.939| 0.749| 0.759|
> | 1.2| 0.928| 0.742| 0.748|
> | A (T:0.8,A:0.4,Al:1.26)| 0.932| 0.743| 0.744|
> | B (T:1.0,A:0.6,Al:0.8)| **0.939**| 0.760| 0.762|
>
> *Tab.4: Sigma Exp. (PLCC)*
>
> **6. T2I with other models**
>
> Following the same setup as in our paper (Sec. 4), we compare reward models under the same protocol.
>
> | Reward Model| Best-of-N| BoN + Revise|
> |-|-:|-:|
> | ImageReward| 0.74| -|
> | UnifiedReward| 0.77| 0.78|
> | **Ours**| 0.78| **0.80**|
>
> *Tab.5: Geneval Score*
>
> VisualScore achieves the best performance. ImageReward outputs only a scalar score without interpretable feedback, so it cannot support interactive revision ("-"). Our reasoning-based scoring provides dimension-specific explanations enabling effective iterative refinement.
>
> **7. Limitations**
> - **Scalability:** We agree this is important. Our pipeline is model-agnostic and expected to generalize. Due to the limited rebuttal timeline, 32B/72B training could not be completed. We will include these in the updated version. We also validated VisualScore on a stronger text-to-image generator, **Flux.2-klein-9B** (Tab.6).
> - **Fixed Template:** The analysis plan is not a fixed template. We generate multiple trajectories per sample varying in structure, order, and emphasis. The model learns flexible reasoning patterns from data. See Appendix Fig. 4–6.
> - **GRPO with 16:** Both 8 and 16 are common sample sizes in RL. N=16 is only marginally better than N=8 (Tab.7), so practitioners can flexibly choose based on their budget.
>
> | | Perf.|
> |-|-:|
> | Baseline| 0.82|
> | BoN by VisualScore| 0.88|
>
> *Tab.6: Geneval w/ Flux.2-klein-9B*
>
> | N| KonIQ|
> |-|-:|
> | 8| 0.935|
> | 16| 0.937|
>
> *Tab.7: Sample Size N (PLCC)*

---

### Decision · Program_Chairs · 2026-04-30

**Decision:**

Accept (regular)

**Comment:**

Initially, the paper received mixed scores, but after the careful rebuttal and follow-up discussion, I find that most of the  technical concerns were substantially addressed. In particular, the authors added the previously missing single-task vs. multi-task comparison, which supports the core claim that naive joint training is harmful while the proposed training strategy makes unified multi-task learning effective. They also provided a Gemini-based experiments, expanded the sigma study, strengthened the T2I reranking comparison with stronger same-protocol baselines, and added further evidence on transfer and scaling, including larger-generator results and a 32B model result that was strong enough to move one reviewer from Weak Reject to Weak Accept.

The remaining disagreement seems to be on how highly one values the paper’s novelty, not on unresolved correctness or empirical support. On that point, I agree that some components are adaptations rather than wholly new mechanisms. However, I do not think the post-rebuttal record supports a view that the method is under-validated or that the main claims are unsubstantiated. One reviewer explicitly stated that the concerns were fully resolved and kept an positive score, another reviewer raised the score to Weak Accept after the additional scaling evidence, and even one of the remaining Weak Reject reviews explicitly acknowledged that the key added experiments were convincing.

Overall, I view this paper as a solid and useful contribution. While I understand the remaining novelty concerns, my overall assessment is that the paper is worth sharing in our ICML community, especially in light of the strengthened post-rebuttal evidence. I would therefore lean to Accept.